# A Review of the Association between Exposure to Flame Retardants and Thyroid Function

**DOI:** 10.3390/biomedicines12061365

**Published:** 2024-06-19

**Authors:** Brandon Yeshoua, Horacio Romero Castillo, Mathilda Monaghan, Maaike van Gerwen

**Affiliations:** 1Department of Otolaryngology-Head and Neck Surgery, Icahn School of Medicine at Mount Sinai, New York, NY 10029, USA; brandon.yeshoua@icahn.mssm.edu (B.Y.); horacio.romerocastillo@icahn.mssm.edu (H.R.C.); mathilda.monaghan@mountsinai.org (M.M.); 2Institute for Translational Epidemiology, Icahn School of Medicine at Mount Sinai, New York, NY 10029, USA

**Keywords:** flame retardant, thyroid hormone, PBDEs, thyroid function, thyroid cancer

## Abstract

Flame retardants have been shown to cause widespread physiological effects, in particular on endocrine organs such as the thyroid. This review aims to provide an overview of the literature on the association between flame retardants and thyroid function within humans. A search in the National Library of Medicine and National Institutes of Health PubMed database through January 2024 yielded 61 studies that met the inclusion criteria. The most frequently analyzed flame retardants across all thyroid hormones were polybrominated diphenyl ethers (PBDEs), in particular BDE-47 and BDE-99. Ten studies demonstrated exclusively positive associations between flame retardants and thyroid stimulating hormone (TSH). Six studies demonstrated exclusively negative associations between flame retardants and TSH. Twelve studies demonstrated exclusively positive associations for total triiodothyronine (tT3) and total thyroxine (tT4). Five and eight studies demonstrated exclusively negative associations between flame retardants and these same thyroid hormones, respectively. The effect of flame retardants on thyroid hormones is heterogeneous; however, the long-term impact warrants further investigation. Vulnerable populations, including indigenous people, individuals working at e-waste sites, firefighters, and individuals within certain age groups, such as children and elderly, are especially critical to be informed of risk of exposure.

## 1. Introduction

The incidence of both autoimmune thyroid diseases and thyroid cancer has been rising significantly over the past several decades [1,2]. This clear increase in incidence warrants a closer examination of potential causes that predispose to thyroid dysfunction, thyroid cancer, and autoimmune thyroid diseases. A potential cause that has been investigated is exposure to environmental pollutants, including flame retardants [3].

Flame retardants are chemicals applied to various materials to help prevent the spread of a fire. They are often used for household and industrial appliances (e.g., furniture, electronics), disbursed within building sites, and in transportation products such as cars and airplanes [4]. These chemicals have also been applied to the protective gear of firefighters to help mitigate the spread of fire [5]. Flame retardants have also been found in human foods and feces, as well as in human serum and urine samples [6].

Flame retardants have been shown to cause widespread physiological effects, including permanent neurological, endocrine, and reproductive toxicity [4]. Flame retardants share structural similarities with thyroid hormones, allowing them to modulate various functions associated with thyroid hormones [2,7,8]. A figure created by the Mount Sinai Library demonstrates the structural similarity between these molecules in Figure 1 below.

Flame retardants are generally classified into several different categories including halogenated flame retardants, which includes brominated flame retardants such as polybrominated diphenyl ethers (PBDEs), hexabromocyclododecane (HBCDs), and tetrabromobisphenol A (TBBPA), and chlorinated flame retardants such as Tris(2,3-dibromopropyl) phosphate (TDBPP or Tris), chlorinated tris (Tris(1,3-dichloro-2-propyl) phosphate). Addition categories include organophosphorus flame retardants including Triphenyl phosphate (TPP), Tris(2-chloroethyl) phosphate (TCEP), Tris(2-butoxyethyl) phosphate (TBEP), as well as other categories including nitrogen-based, inorganic, intumescent, mineral, and reactive flame retardants [4].

The rising health concerns and regulatory response of flame retardants and their implications on human health have significantly changed overtime. While these molecules have been around for centuries, there has been a significant rise in their use in the 20th century [4]. In 1953, the United States passed the Flammable Fabrics Act, which required the use of flame retardants in many children’s products like clothing and interior furnishings like carpets and rugs [9]. In 1975, California legislators passed the flammability standard, TB-117, which required more stringent implementation of flammability standards to ensure furniture safety from hazards of ignition [10]. However, in 1977, the earliest flame retardants, polychlorinated biphenyls (PCBs), were banned in the United States for their significant toxic effects on endocrine disruption, cancer, liver damage, and neurodevelopmental abnormalities. Industries switched from chlorinated flame retardants to the use of brominated flame retardants [10]. After noting the toxicity and persistence of these effects, with additional recognition of reproductive implications and neurobehavioral effects on children, the European Union took further measures to ban several PBDEs in 2008 [6].

More recently, several countries have taken measures to ban exposure to flame retardants. Countries in the European Union spearheaded efforts to ban the use of brominated flame retardants within electronics, furniture, and other products due to a combination of environmental concerns from persistence and permeability in the environment and human health effects such as endocrine disruption, neurological interference, reproductive issues, and potential carcinogenic properties [11]. The United States Consumer Product Safety Commission has also banned the sales of certain products that contain PBDEs for similar reasons as prior legislation with a particular emphasis on vulnerable populations including infants and children [12]. Many flame retardants, including tetrabrominated diphenyl ethers, hexabrominated diphenyl ethers, decabromodiphenyl ether, and hexabromocyclododecane, have been listed under several legislations, including the Stockholm Convention on Persistent Organic Pollutants [13]. Several states within the United States, including New York and California, have begun passing bills against products containing these molecules [14]. New York became the first state in the nation to restrict the use of halogenated flame retardants in 2021 within electronics. Washington State has also released reports on the detriments of certain flame retardants to vulnerable populations, including firefighters [14]. The Safer States organization released a statement titled “Sign on Letter to Textile Certifiers” addressing polybrominated and polymeric flame retardants that substantiated concerns about the continued use of flame retardants. This bill served as a catalyst for enacting stricter legislation, with a recommendation to expand analyses of preexisting flammability standards to a broader set of products [14].

Although most research investigating the impact of flame retardants on the thyroid has been conducted in animal and preclinical models, several studies analyzed this association in humans. An example is the study by Hoffman et al. who reported that patients exposed to higher concentrations of the flame retardant brominated diphenyl ether 209 (BDE-209) were 2.25 times more likely to develop papillary thyroid cancer compared to matched controls with low BDE levels [2]. In 2015, a meta-analysis including 16 human studies assessing the association between polybrominated diphenyl ethers (PBDEs) and serum thyroid hormone demonstrated specific concentration cut-off values of PBDEs in either serum or cord blood where those below 30 ng/mL had negative correlations with thyroid hormones, those between 30 and 100 ng/mL did not demonstrate a correlation, and those >100 ng/mL demonstrated positive associations with thyroid hormone concentration [15]. This study hypothesized that the heterogeneity in prior studies may be a result of two separate associations depending on acute exposure, where the relationship between PBDEs and thyroid hormones is mono-phasic versus chronic exposure, where the association is hypothesized to be a u-shaped relationship [15].

Other human studies analyzed flame retardant exposure in specific populations, such as pregnant participants and their fetuses [16,17,18,19,20,21,22,23,24,25,26]. An example is the Health Outcomes and Measures of the Environment (HOME) study, a prospective birth study of 389 pregnant participants collecting maternal PBDE concentrations and measuring the effect on maternal serum and cord thyroid hormone concentrations [23]. This study demonstrated specific positive associations between certain flame retardants (e.g., BDE-28 and BDE-47) with maternal thyroid hormone concentrations but no correlations with changes in cord blood. A prospective study in South Korea found similar associations between BDE-47 and BDE-99 with TSH concentration in infant cord blood and bloodspot, respectively [20].

This review not only examines the relationship between flame retardants and thyroid hormone levels, but also analyzes their relationship with cancer.

## 2. Materials and Method

### 2.1. Search Strategy and Selection Criteria

Studies investigating the association between various flame retardants and thyroid hormone levels in humans were identified in the National Library of Medicine and National Institutes of Health PubMed database through January 2024. The databases were searched using the following terms: “Thyroid Gland” [Mesh] OR “thyroid” OR “Triiodothyronine” [Mesh] OR “triiodothyronine” OR “Thyroxine” [Mesh] OR “thyroxine” AND “Flame Retardants” [Mesh] OR “flame retardant*” OR “fire retardant*” OR “Polybrominated Biphenyls” [Mesh] OR “polybrominated biphenyl*” OR “Halogenated Diphenyl Ethers” [Mesh] OR “PBDE” OR “BDE” OR “bromodiphenyl ether”.

The various flame retardants included in the study were polybrominated diphenyl ethers (PBDEs), polybrominated biphenyls (PBBs), triphenyl phosphate (TPHP), tris(1,3-dichloro-2-propyl) phosphate (TDCIPP), tris(2-chloro-1-methylethyl) phosphate (TCIPP), tris(2-chloroethyl) phosphate (TCEP), polychlorinated dibenzodioxins (PCDDs) and dibenzofurans (PCDFs), decabromodiphenyl ethane (DBDPE), tetrabromobisphenol A (TBBPA), bis-2-chloroethyl phosphate (BCEP), and less commonly analyzed flame retardants such as hexabromocyclododecane (HBCD), dibutyl phosphate (DBuP), and di-p-cresyl phosphate (DpCP).

Articles were uploaded into Covidence systematic review software (https://www.covidence.org/), Veritas Health Innovation, Melbourne, Australia, a web-based collaboration software platform that streamlines the production of systematic and other literature reviews for the removal of duplicate articles and further inclusion assessment. Articles were initially screened for relevance based on the title and abstract by two independent reviewers (B.Y., H.R.), followed by a full-text review (B.Y., H.R.). Any disagreements between the two reviewers were resolved through discussion with a third reviewer (M.v.G.) until a consensus was reached.

Articles were included if they met the following inclusion criteria: (1) provided measurements of serum, plasma, cord, placental, breast milk, or urine flame retardants, (2) provided measurements of serum, plasma, cord, placental, breast milk, or urine thyroid hormones (free thyroxine (fT4), total thyroxine (tT4), free triiodothyronine (fT3), total triiodothyronine (tT3), or thyroid-stimulating hormone (TSH)), (3) investigated the association between flame retardants and thyroid hormones within humans, or (4) investigated the association between flame retardants with thyroid-related antibodies or (5) cancer. Articles were excluded for the following reasons: (1) systematic reviews or meta-analyses, (2) non-human studies, (3) in vitro and preclinical studies, (4) missing statistical analysis, (5) studies not including flame retardants, and (6) studies without statistical analyses. Articles with incorrect or unspecified outcome variables included measurements of quantities of metals such as lead excretion or enzyme function like thyroid deiodinase activity as primary outcome variables were also excluded.

### 2.2. Data Extraction and Analysis

The information extracted from the eligible studies included the study type, year of publication, country in which the study was conducted, study population, type of flame retardant, measurement of exposure, number of flame retardants, and the association with thyroid function. The information was extracted and compared by two researchers (B.Y. and H.C.). Populations excluding individuals < 18 years of age were classified as “adults”. Studies were classified as “prenatal” if the study included analyses of newborns with or without their mothers. Data were extracted on the specific effects of the association between different flame retardants and thyroid hormones, including TSH, T3, T4, fT3, fT4, thyroid cancer, and autoimmune conditions (including thyroid-related antibodies). The data gathered on anti-thyroid autoantibodies and thyroid cancer generally examined a population of age- and/or gender-matched healthy patients to a population of patients diagnosed with thyroid cancer. The measurements of flame retardants are often lipid-adjusted to account for the lipophilic qualities of these chemicals and standardized for differences in individual fat content. Several studies identified differences in the risk of cancer development based on the concentration of flame retardants [2,27,28]. Statistical analyses, most commonly logistic regression and odds ratios, were used to establish this association. Patients with a greater exposure to certain flame retardants were noted to have higher odds (OR > 1) of developing thyroid cancer. Sex-specific data were extracted where available. If a study included an association for individual flame retardants as well as sums of flame retardants, all information was extracted. Studies that met the inclusion criteria but also reported associations with excluded compounds (e.g., PCBs, dioxins, per- and polyfluoroalkyl substances (PFAS)) were included but only the results of flame retardants were reported.

## 3. Results

This initial review yielded 772 studies, of which 61 studies met the inclusion criteria and were included in the review. The search and selection process is described in Figure 2.

There were 31 cohort studies, 20 case-control studies, and 10 cross-sectional studies (Table 1). The year of publication ranged from 1980 to 2023. The most frequently analyzed flame retardants in the studies were PBDEs, with most studies performed in adult populations. Additional groups included evaluation in newborns to pregnant participants and school children living near petrochemical complexes [29,30]. Additionally, several studies analyzed patients with preexisting thyroid cancer [2,27,28,31]. Most studies (42 out of 61 studies (69%)) were published at or after the most recently published meta-analysis by Zhao et al. in 2015 [15], with only 6 studies meeting the inclusion criteria before 2010 [32,33,34,35,36,37]. Most studies were conducted in the United States (n = 26), followed by China (n = 19) (Table 1).

### 3.1. Flame Retardants and Thyroid Function

#### 3.1.1. Flame Retardants and TSH

A total of twenty-five studies analyzed the associations between flame retardants and TSH levels (Table 2). Ten of the twenty-five studies demonstrated positive associations between all flame retardants and TSH [20,24,29,31,32,34,35,39,51,63,71,73]. Six studies showed negative associations [17,37,52,54,56,59]. Two studies showed both positive and negative associations [26,70], and seven studies revealed no associations [18,38,41,46,51,71,77].

Among the ten studies that exclusively demonstrated positive associations, BDE-47 and BDE-99 were the most frequently analyzed flame retardants and were found to be positively associated with TSH in five [29,34,63,70,73], and three studies [20,35,63], respectively (Table 2).

Of the six studies that showed exclusively negative associations, BDE-47 was the most frequent flame retardant analyzed, with five studies showing negative associations with the specific flame retardant (Table 2). Two studies showed a negative association with BDE-100 [17,54]. Special populations included pregnant women in New York City (NYC) and their newborns [52], mother–child pairs [59], and pregnant women in California [17].

The studies that yielded both positive and negative associations between flame retardants and TSH were conducted in diverse settings. These included a newborn population in Korea [20], pregnant women in the United States [26], and patients with and without thyroid cancer [41], as well as two rural coastal populations in Canada [38] (Table 2).

#### 3.1.2. Flame Retardants and tT3

Twenty-six studies examined the association between flame retardants and tT3 levels. Of these, twelve revealed exclusively positive associations [21,23,30,33,35,45,50,51,54,61,71,74] (Table 3). Seven studies demonstrated negative associations [16,29,43,65,72,73,76]. A total of five studies resulted in mixed positive and negative associations between flame retardants and tT3 [41,44,49,70,75]. Two studies found no association between flame retardants and tT3 [46,68].

BDE-47 and BDE-99 were the most frequent flame retardants showing positive associations with tT3. Six and four studies showed positive associations with BDE-47 [23,30,33,45,54,61] and BDE-99, respectively (Table 3). Notably, four studies showed negative associations with BDE-47 [16,49,70,72] and three studies showed negative associations with BDE-99 [16,58,75].

Abdelouahab et al. analyzed a population of 397 pregnant women from the University Hospital Center of Sherbrooke in Quebec, and they found that two of the individual PBDEs were negatively associated with thyroid hormone concentrations, including tT3 levels in their lipid-based model at the time of prenatal visit and in both their lipid-based model and volume-based model at the time of delivery [16]. Several other studies found similar correlations but within different populations. Huang et al. analyzed a group of volunteers from the provinces of Shanxi and Liaoning in northern China for specific PBDEs and their association with thyroid hormones. They found negative associations between BDE-17, BDE-28, BDE-47, BDE-153, and BDE-183 and concentrations of tT3 [70] (Table 3). Similarly, Kim and colleagues found a significant negative association between BDE-47 and tT3 in pregnant women in Korea [72]. Li and colleagues also found a similar negative association between BDE-47, BDE-99, BDE-100, BDE-197, BDE-203, BDE-207, and tT3 in a cohort of women from Germany in 2015–2016 [49].

Most of the studies analyzing the association between flame retardants and tT3 specified that the location of the individuals affected played a significant role in the association between flame retardants and their thyroid levels. Studies analyzing individuals who work at an e-waste site were noted to have positive associations with tT3 [28,44,54,78]. Yet, other studies were conducted in young children and indigenous populations. Guo et al. analyzed a population of fifth graders from China for the association between BDE-209 and tT3 concentrations in their serum [30]. They found a positive association between BDE-47 and tT3 concentrations (β = 0.080, *p* < 0.05, 95% CI: 0.011, 0.15). Kim et al. analyzed the association between various flame retardants in children with congenital hypothyroidism [73]. They found a similar statistically significant inverse association between BDE-153 and tT3 in this population of patients [73]. Dallaire et al. investigated Inuit adults in Canada, finding a positive association between BDE-47 and tT3 [33]. BDE-153 demonstrated strong negative associations with tT3 concentration in all five studies that analyzed the association with the tT3 concentration [29,43,65,70,73]. 

Five of the twenty-six studies reported positive and negative associations between flame retardants and tT3 [41,44,49,70,75]. Liu et al. analyzed the serum samples in Chinese patients with and without thyroid cancer and noted a non-significant negative association with decabromodiphenyl ethane (DBDPE) [41]. Other flame retardants in the same study including triphenyl phosphate (TPP) and ethylhexyldiphenyl phosphate (EHDP) demonstrated statistically significant positive associations. Zhao et al. studied residents of a BFR-producing region in China, and they found positive associations between PBEB and tT3, but negative associations between DBDPE and tT3 [44]. Huang et al. investigated the relationship between tT3 and various flame retardants in a cohort of volunteers from northern China. They found significant negative associations between BDE-17, BDE-28, BDE-47, BDE-153, and BDE-183 with tT3, but they found statistically significant positive associations between BDE-99 and BDE-208 and tT3 [70].

#### 3.1.3. Flame Retardants and tT4

The association between flame retardants and tT4 concentrations was analyzed in twenty-three studies. Twelve of the twenty-three studies found positive associations between flame retardants and tT4 levels [19,22,23,34,35,37,43,51,57,60,61,76] (Table 4). Nine studies found negative associations between flame retardants and tT4 levels [16,30,31,40,49,54,58,62,67]. One study found no associations between flame retardants and tT4 [41]. One study found mixed associations between flame retardants and tT4 levels [46]. 

The most analyzed BDEs among this cohort were BDE-47, BDE-99, and BDE-153, all demonstrating mixed results. Among studies with positive associations, several associations were noted between specific flame retardants and tT4 levels. BDE-47 was noted to have positive correlations with tT4 in one study [23]. Stapleton et al. [22] and Vuong et al. [23] demonstrated positive associations in populations of pregnant women where measures were taken from serum samples and/or cord samples (rs = 0.20, *p* < 0.05; β = 0.82, *p* < 0.05, 95% CI: 0.12, 1.51). Additionally, some less commonly analyzed flame retardants, including DBDPE and PFOs, were also noted to have a positive association with tT4 concentration [51,60,79]. DPHP was noted to be positive in some studies and negative in others [46]. BDE-153 was noted to have a positive association in three studies [22,34,43]. Yang et al. established a positive relationship between BDE-153 and tT4 levels in patients with preexisting abnormal thyroid hormone levels (β = 1.11, *p* < 0.05, 95% CI: −0.1, 2.23) [43]. Only one study demonstrated a positive association between BDE-99 and tT4 concentration [19].

Within the studies analyzing negative associations between the most frequent flame retardants, several had notable associations. BDE-47 demonstrated negative correlations with tT4 in four separate studies [16,30,31,67]. Most studies analyzing the association between BDE-99 and tT4 also demonstrated a negative association [16,49,58,62]. BDE-153 demonstrated a negative association in two studies [54,62]. Still, other studies demonstrated negative associations between flame retardants and tT4 concentration. A recent study by Trowbridge et al. analyzed the association of BDCPP with tT4 concentration in urine samples of female firefighters [40]. This cross-sectional study compared the association of flame retardants with thyroid hormones in firefighters to office workers. The median levels of flame retardant BDCPP analyzed were five times higher in the firefighter cohort. A separate study looked at the association between placental levels of various persistent organic pollutants in the mothers of boys with and without cryptorchidism, which found that BDE-99, BDE-100, and the sum of BDE-47, BDE-99, and BDE-100 were negatively associated with tT4 concentration [58]. A separate study analyzed this association in preexisting thyroid cancer patients [31]. The authors found a negative association between BDE-47 and tT4 (β = −2.49, 95% CI: −4.19, −0.78).

#### 3.1.4. Flame Retardants and fT3

Nineteen studies analyzed the association between flame retardants and fT3 levels (Table 5). Certain flame retardants, particularly BDE-28, BDE-47, and BDE-99, were most commonly associated with fT3. Eight of the nineteen studies found positive associations between flame retardants and fT3 levels [16,23,29,45,50,53,58,59]. One study found positive associations between BDE-28 and fT3 levels [23]. BDE-47 was found to have a positive association in three studies [23,44,54]. BDE-99 was most notably positively associated with fT3 levels in two separate studies [16,44]. 

Four of the nineteen studies demonstrated negative associations between flame retardants and fT3 concentrations [42,72,75,76]. BDE-99 was negatively associated with fT3 levels in a single paper [75]. Only one study found a negative association between BDE-28 and fT3 levels [42]. Some less commonly analyzed flame retardants, including PBB-103 and BDCIPP, also demonstrated negative associations with fT3 levels [21,76].

Five of the nineteen studies demonstrated mixed associations between flame retardants and fT3 levels [21,38,41,46,54]. Two of the eighteen demonstrated no association between flame retardants and fT3 [32,68].

#### 3.1.5. Flame Retardants and fT4

The association between flame retardants and fT4 concentrations was analyzed in twenty-five studies (Table 6). The most investigated flame retardants included BDE-47, BDE-99, and BDE-153. Fourteen of the twenty-four studies found positive associations between the flame retardants and fT4 concentrations [16,21,22,23,33,37,51,56,61,67,71,72,73,74]. Three studies found positive associations between BDE-47 and fT4 concentrations [16,22,23]. In studies where BDE-99 was analyzed, three found positive associations [16,22,74]. In studies where BDE-153 was analyzed, five found positive associations [22,34,38,67,73] (Table 6).

Only six of the twenty-three studies demonstrated negative associations between the analyzed flame retardants and fT4 concentration [30,42,43,54,75,76] (Table 6). Albert et al. investigated a sample of 153 men from Montreal, Canada, and found that BDE-47 is negatively associated with levels of fT4 [56]. Three studies found negative associations between BDE-47 and fT4 concentration [30,42,43]. Three studies found negative associations between BDE-99 and fT4 [30,42,75]. Only one study found a negative association between BDE-153 and fT4 [54].

One study found mixed results between flame retardants and fT4 levels [41]. Four studies found no association between flame retardants and fT4 [32,38,46,68].

#### 3.1.6. Flame Retardants and Thyroid Cancer

Seven case–control studies analyzed the association between flame retardants and the risk of thyroid cancer in humans (Table 7). Four studies found a positive association between flame retardant exposure and the risk of thyroid cancers [2,28,47,48]. Huang et al. examined a population of active US military personnel and noted that BDE-28 was associated with a significantly increased risk of the classical form of papillary thyroid carcinoma (OR = 2.09, 95% CI: 1.05, 4.15, *p* = 0.02) [47]. This study noted that this association was more pronounced in females than males and was tumor-size dependent. Deziel and colleagues found an inverse relationship between BDE-209 concentration and the risk of PTC [27].

One study identified mixed associations between flame retardants and thyroid cancer [41]. Within this study, patients with and without thyroid cancer were analyzed for their associations with several different flame retardants, including PBB, PBT, HBB, EHTBB, BTBPE, DBDPE, TPrP, TBP, TCEP, TCPP, TDCPP, TBEP, TPP, and EHDP. All flame retardants except HBB, DBTPE, and TPP demonstrated positive associations with thyroid cancer [41] (Table 7).

A single study found no association between flame retardants and the risk of thyroid cancer [66]. This study investigated the relationship between BDE-47, BDE-99, BDE-100, and BDE-153 and the association with both papillary thyroid cancer and all types of thyroid cancer but found no association in either cohort (Table 7).

#### 3.1.7. Flame Retardants and Thyroid-Related Antibodies

Two studies have noted the association between flame retardants and elevated thyroid-related antibody concentrations [32,51] (Table 8). They found correlations with thyroid peroxidase antibodies (anti-TPO antibody), an antibody often associated with Hashimoto thyroiditis and postpartum thyroiditis [44,51]. Two studies reported positive associations between flame retardants and anti-TPO antibodies [44,51]. Chen and colleagues analyzed the flame retardant DBDPE in adults within a manufacturing area in China. This same study noted a correlation between DBDPE and a separate antibody, thyroglobulin antibody (anti-TG), which was reported to have a positive association [44,51]. No studies found negative associations between flame retardants and TPO antibodies (Table 8).

## 4. Discussion

The known impact of flame retardants on the thyroid highlights the importance of continued research on this topic. The number of studies published since the most recently published systematic review in 2015 and the heterogeneity of contemporary research warranted a renewed synthesis to help in drawing potential conclusions regarding the impact of flame retardants on thyroid disease. This review reconfirmed that flame retardants have a very heterogeneous impact on the thyroid, depending on the type of flame retardant and thyroid hormones; however, multiple flame retardants are associated with thyroid disruption and even thyroid cancer.

One recent meta-analysis by Van der Schyff and colleagues examining the prevalence of flame retardants demonstrated longitudinal changes in the concentration of flame retardants before and after legislative intervention in 2013 globally [80]. They analyzed the concentration within breast milk immediately after the implementation of regulations banning the use of certain flame retardants [80]. A significant decrease in BDE-47 and BDE-99 was reported in Europe around the time the regulation was implemented [80].

While the associations between flame retardants and thyroid hormones are variable and dependent on a multitude of factors, the relationship between certain flame retardants and thyroid hormones demonstrates stronger correlations. Certain flame retardants including BDE-47, BDE-99, BDE-100, and their association with thyroid hormones have been more prominently described in the literature. The hydrophobic structure of these molecules and their similar chemical configuration as thyroid hormones explain several of the proposed mechanisms involved in their ability to dysregulate thyroid function, which include interactions with signaling pathways and cellular membranes, damage to DNA and alterations to gene expression, and adjustments to the cell cycle and cell death [8,81]. Most flame retardants are not chemically bound to their resins, allowing them to freely dissociate and be ingested, inhaled, or transmitted through diet and across the placenta to the fetus [2]. T4 and T3 share the greatest structural similarities with various flame retardants, explaining their endocrine-disrupting effects [2]. Several studies have proposed competitive inhibition to the transport molecules of thyroid hormones including thyroid binding globulin (TBG) and transthyretin and upregulation of glucuronidase, an enzyme involved in the clearance of thyroid hormones [46,82,83]. Gravel et al. noted that the hydroxylated metabolites act as competitive inhibitors that ultimately prevent gene expression [46]. This paper also noted that the allosteric activation of several organophosphate esters, TPhP and TDCIPP, has been observed (GRAVEL). These molecules would increase the binding of free T4 to transport proteins through this mechanism, resulting in conformational changes in transport proteins [46]. An additional newer proposed mechanism of more recent flame retardants, including DBDPE, involved inhibiting thyroid deiodinase [84]. Several papers also noted this as a mechanism of PBDEs inhibiting the activity of sulfotransferases, enzymes involved in the metabolism of thyroid hormones [22,64,67]. Moreover, Makey et al. noted that PBDEs result in a direct reduction in tT4 levels, and that these effects may be tissue-specific [67]. Leonetti et al. also described how brominated flame retardants inhibit deiodinases, specifically DIO3, within the placenta [64]. This paper also described how flame retardants may influence sulfotransferase activity in the placenta [64]. Hormone ratios, including fT4 to tT4 and fT3 to tT3, can be used as indicators of transport protein involvement, whereas the ratio of fT4 to fT3 can be used as an indicator of deiodinase involvement, an enzyme that is used in the conversion of T4 to T3 [46]. Similarly, a mechanism of action associated with DBDPE is the organ-specific accumulation of DBDPE within the liver, causing an increase in hepatic detoxification enzyme function including CYP and UDGPT. This results in greater metabolism of the flame retardants and conversion of T4 to T3 [51]. Other studies have analyzed the effects of more recent flame retardants such as BTBPE and their effects on sodium/iodine symporter (NIS) levels [25]. Two additional studies have noted intranuclear mechanisms of action from several PBDEs and metabolites acting on receptors such as thyroid hormone receptors and estrogen receptors, modulating the transcription of several genes [26,71]. Zhang et al. found that BTBPE levels reduced protein levels of PAX8, TTF1, and TTF2, inhibiting thyroglobulin and NIS [25]. More recent studies have begun analyzing the molecular relationships between specific flame retardants and thyroid hormones, including analysis of binding energies [13]. Sheikh et al. analyzed the structural binding of PBDEs against the ligand binding pocket of thyroid receptor alpha and identified the specific amino acid residues and quantified the interactions in the binding pocket [13]. Within human hepatic tissue, PBDEs are converted into their more active metabolite, OH-PBDEs, a chemical with even greater structural similarity to thyroid hormones. These hydroxylated PBDEs have been found to possess stronger endocrine disruption through protein binding to molecules such as transthyretin and TBG with higher affinity resulting in a shorter half-life of T4, in addition to the inhibition of thyroid deiodinase [26,82], and interfere with the binding of thyroid hormones to human receptors [76]. This reaction may also explain why the analysis of flame retardants in vitro and in animal studies may differ from analyses within humans. Yet further unelucidated mechanisms are suspected, as the relationship between flame retardant exposure and carcinogenicity is thought to be regulated by additional indirect and chronic mechanisms [47].

The techniques used to effectively measure flame retardants varied across different studies. Most studies analyzed either human serum or urine samples to measure the concentrations of flame retardants. Yet, other studies looked at breast milk, cord serum, cord blood, or even hair and nails samples. Wang et al. used wristbands, a highly effective and low-effort intervention that may help to evaluate areas with high flame-retardant concentrations [55].

While definitive conclusions on all the associations between flame retardants and thyroid hormone regulation and thyroid cancer are heterogenous and not yet elucidated, several meaningful conclusions can be drawn from the literature. The mixed associations between many similar flame retardants and thyroid hormones indicate that additional factors must be at play. Several considerations noted in papers include variables such as age, BMI, the level of exposure, and legislation in each country or region. Similarly, pregnant women produce greater levels of thyroid hormones to help support the development of the mother and the fetus. Thus, disruption of thyroid hormones is particularly detrimental in this population. Additionally, elderly patients are noted to be at greater risk than the average population as well. The accumulation of these molecules over time with the corresponding decline in thyroid function in aging individuals subjects these patients to greater detrimental effects of flame retardants [38,53,69].

The more profound effects of exposure to these toxic molecules also depend on the level of exposure, often in an occupational setting. Chen and colleagues noted a significant increase in the level of exposure of DBDPE in manufacturing workers compared to non-working exposed residents [51]. Other vulnerable populations such as children in schools may inadvertently be affected. Guo and colleagues [30] noted that fifth-grade students in one of two schools living near a petrochemical complex had greater exposure and more downstream thyroid disrupting effects from these molecules [30].

Additionally, a greater concentration of flame retardant exposure is associated with a greater dysregulation of thyroid hormones and increased risk of developing thyroid cancer [2,27,28,41,47,48,66]. Huang and colleagues [47] found that higher concentrations of BDE-28 are associated with a higher risk of large classical PTC, which is especially notable within women [47]. A similar pattern was noted by Kassotis and colleagues through a separate mechanism of their wristbands. They found that higher concentrations of flame retardants, such as TDCIPP, were also associated with increased toxicity [48]. Li and colleagues also noted that greater concentrations of PBDEs in their study correlated with greater adverse health outcomes [49]. They noted specific adverse outcomes based on certain flame retardants, including BDE-99 with specific alterations in thyroid hormone levels during childhood and BDE-154 with significant changes in the thyroid hormone level in both maternal and cord blood [49]. Moreover, Hoffman et al. noted that different flame retardants are correlated with different forms of cancer. Specifically, BDE-209 was found to be associated with smaller, less aggressive forms of PTC, whereas TCEP was associated with larger, more aggressive forms of PTC that had extrathyroidal extension and nodal metastasis [2].

Additionally, location, whether it be country, state, or even region-specific differences in both the concentrations and regulations of flame retardants, represents a notable consideration. Jacobson et al. noted that the concentrations of PBDEs analyzed in their study were similar to those of other studies conducted in the US, an average concentration 2–3× higher than that noted in other locations like Europe and Asia [63]. Similarly, Lignell et al. noted that the concentration in their population of Swedish pregnant women was about ten times lower regarding body burden than in comparable studies in North American women [65]. Zhang and colleagues analyzed legislative intervention for the flame retardant BDE-209, found in relatively higher concentrations than other flame retardants both in the environment and in humans. This study, conducted in China, noted relatively higher concentrations than in the studies conducted in the United States [2,27,64], likely due to the lack of regulation on this flame retardant at the time of publication. Similar extremely high relative concentration and detection frequencies were noted by Zhao and colleagues [50]. Only recently has China included BDE-209 in their list of contaminants under the List of Key Emerging Contaminants under Control in 2023 [85].

Moreover, vulnerable populations, including indigenous people, individuals within certain occupations, such as those working at e-waste sites, firefighters, coastal inhabitants, and individuals within certain age groups, such as children and the elderly, must be assessed for exposure with precaution as these populations have demonstrated a greater vulnerability [18,40,53,54,61,68,73,74,76]. Infants and children represent a population in which thyroid hormone regulation is especially critical in development. Vuong et al. noted that prenatal exposure to certain flame retardants including BDE-47, -99, and -100 predisposes children not only to thyroid dysregulation but also to cognitive impairment and ADHD-like behavior [59]. Interference with regulation by flame retardants has an especially serious effect on this population. Not only is their exposure to these molecules relatively more dangerous than the general population, but children are also likely to be exposed to greater quantities of flame retardants. These molecules, present in dust, furniture, and electronics, are easily ingested by children. The smaller body size and higher metabolic demands of children also subjects them to greater consumption, which proportionally contains greater levels of these toxins [4].

Moreover, women are at a greater risk than men for the development of thyroid dysregulation and cancer [27,47]. Preston et al. [60] noted that the higher concentrations of DPHP have more profound effects on thyroid hormone levels in women compared to men [27,47,55,60,66]. The proposed reasoning behind these sex-specific associations is due to increased levels of estrogen, which plays a crucial role in thyroid regulation [26,27,55,60,71]. Additionally, women have a greater sensitivity to thyroid hormones due to more frequent physiological hormone level changes through pregnancy and menstrual cycles. They also have higher rates of thyroid autoimmune conditions and thyroid cancer, making them more susceptible to thyroid disrupting factors [26,27,55,60]. Other factors include an increased body fat percentage, resulting in a greater accumulation of flame retardants [55,60,71]. Trowbridge and colleagues found that the concentrations of bis(1,3-dichloro-2-propyl) phosphate are five times higher in firefighters than in office workers [40]. Additionally, occupation-specific effects of flame-retardant exposure on thyroid hormones were also noted in this population, where exposure to BDCPP led to a reduction in T4 only in a population of firefighters [40]. Additionally, the flame retardant exposure of children, a vulnerable population that may incur long-term effects because exposure occurs at an age where physiological development is still taking place, may have significant long-term consequences on physiological development [18,25,52,53,73,74]. A recent study by Kim et al. also found that specific flame retardants impact the aging population by showing an association with premenopausal and postmenopausal status in females and age over or under 50 in males [77]. In 2021, the European Food Safety Authority commented on HBCDDs, concluding that this molecule does not pose significant health risks except in breastfed infants consuming high quantities of breast milk [11]. More recently, in January 2024, the European Food Safety Authority published increasing research on the detriments of PBDEs within food [11]. They noted that the greatest implications were on the reproductive and nervous systems. Additionally, while a subset of studies found sex-specific associations, there is a need for greater focus on the implications of exposure to women as most of the occupations and thus studies have been analyzed in a male-predominant population [40]. Additionally, the ability of transplacental acquirement of these molecules to the fetus makes this population an increasingly critical group to analyze. Finally, greater attention should be given to the “impact to mixture exposure” when looking at different endocrine disrupters [1], as this may contribute to the complexity of the relationships with various thyroid hormones. Factors such as patient age, body-mass index, occupational exposure, sex, and even health literacy and awareness should be considered when identifying and addressing at-risk populations. For example, the inverse association noted in Deziel and colleagues’ analyses with PTC is not intuitive, yet may be explained by factors including the design of the study, and timing of sample collection, confounding variables that may be correlated with lower BDE-209 levels and a higher risk of cancer, a possible threshold effect inherent within BDE-209 where effects on thyroid function differ based on a certain threshold, or possibly due to statistical artifacts and chance [27]. Additionally, considering the mixed correlation results of many flame retardants, future studies should place greater emphasis on the other variables that may be influencing the lack of consistent associations between flame retardants and thyroid hormones. It is critical to create statistical models that account for variables such as age and gender, BMI, occupational exposure, but also educational awareness and other socioeconomic determinants of health that affect access to proper medical care and thus influence the ability for early detection. It is similarly important to focus on identifying at-risk populations with exposure to flame retardants, educating populations on the potential repercussions of exposure, and implementing strategies to mitigate it. Moreover, future studies may also benefit from more widespread and robust detection of flame retardants within the environment and within humans. As legislation has universally been shown to be slow to adapt, studies may also benefit from surveying whether patients and physicians are aware of the detriments of flame retardants.

This review inherently has several limitations. The varying half-lives and retention time of different flame retardants in the body, as well as the duration of the time that passed before the flame-retardant concentration was measured, differed between studies. Different studies also collected data through different mechanisms including serum, urine, cord blood, cord serum, and even hair and nail samples. This non-standardized approach of measurement affected the ability to standardize the results. The variability in the type of flame retardant, the level of exposure and timing of measurement, the length of exposure, and the population at risk across the studies made comparing the concentration of flame retardants across papers difficult. However, certain papers provided relative concentration comparisons in addition to reporting the detection frequencies of flame retardants. Moreover, the generalizability of results is another limitation, as several studies analyzed specific and often isolated populations. For instance, Dallaire and colleagues analyzed a distinctive population of about 14,000 people living in Nunavik in Northern Canada, the Inuit population [33]. Babichuk and colleagues analyzed two isolated rural population in Canada, screening primarily for locals consuming seafood products [38]. These populations have different exposure risks to flame retardants and are not uniformly representative of the average individual analyzed across all humanized studies.

## 5. Conclusions

This review suggests that although the association between flame retardants and thyroid dysfunction is heterogeneous and dependent on the type of flame retardant and thyroid function, exposure may have serious implications in particular for certain at-risk populations. Future studies should further explore the mechanisms through which these associations present. Moreover, future research analyzing the effects of flame retardants on thyroid hormones may benefit from standardization for known exposure risk within identified populations such as firemen and those located by e-waste sites. Further investigation and publicity are needed to better understand the relationship between exposure to flame retardants and thyroid hormones, which will inform legislative intervention to educate on the potential negative health impact of flame retardants.

## Figures and Tables

**Figure 1 biomedicines-12-01365-f001:**
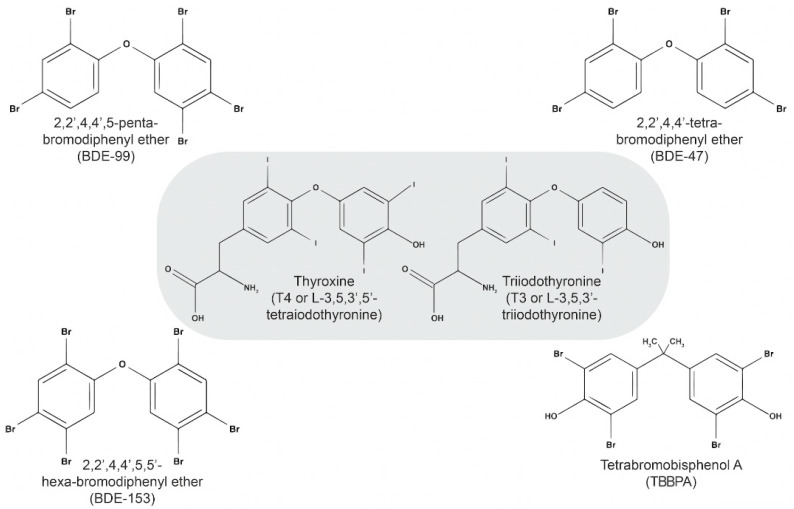
Structure similarity between thyroxine (T4) and triiodothyronine (T3) and 2,2′,4,4′,5-Pentabromodiphenyl ether (BDE-99), 2,2′,4,4′-Tetrabromodiphenyl ether (BDE-47), 2,2′,4,4′,5,5′-Hexabromodiphenyl ether (BDE-153), and tetrabromobisphenol A (TBBPA).

**Figure 2 biomedicines-12-01365-f002:**
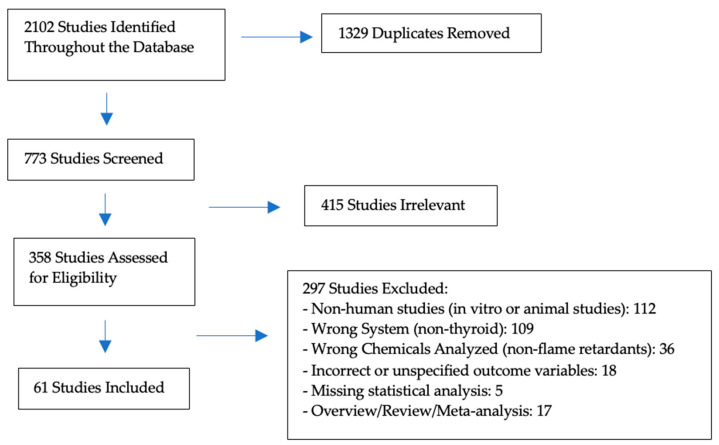
Search and selection strategy.

**Table 1 biomedicines-12-01365-t001:** Overview of included studies.

Author (Year), Ref.	Flame Retardant Type	Country	Study Design	Investigated Population (n)	Measurement of Exposure	Relative Exposure Level; Concentration, Detection Frequency (DF)
Babichuk (2023) [38]	PBB-153, PBDE-28, PBDE-47, PBDE-99, PBDE-100, PBDE-153	Canada	Cohort	Two rural coastal populations (n = 80)	Serum	Not specified; All DFs > 70% PBB-153: 0.57 PBDE-28: 0.00 PBDE-47: 4.84 PBDE-99: 1.63 PBDE-100: 0.00 PBDE-153: 9.03
Liu (2023) [39]	TDCIPP, TEP, and TCEP	China	Cohort	Females of childbearing age (n = 319)	Serum	High TCEP concentration relative to TDCIPP and TEP TDCIPP: 2.03 ng/dL; DF: not specified TEP: 1.09 ng/mL, DF: >50%, TCEP: 0.33 ng/mL, DF: 96.6%
Trowbridge (2022) [40]	BDCPP	USA	Cross-sectional	Female firefighters and office workers from San Francisco (n = 165)	Urine	5× higher in firefighters than office workers Firefighter: 4.08 pmol/g, DF: 100% Office workers GM: 0.90 pmol/g, DF: 90%
Liu (2022) [41]	PBB, PBT, DPTE, HBB, EHTBB, BTBPE, DBDPE, TPrP, TBP, TCEP, TCPP, TDCPP, TBEP, TPP, EHDP	China	Case–control	Patients with or without thyroid cancer (n = 481)	Serum	Not specified PBB: <MDL ng/g/lw; DF: 25% PBT: 0.2 ng/g/lw, DF: 77%, DPTE: 3.7 ng/g/lw, DF: 50% HBB: <MDL, DF: 20% EHTBB: <MDL, DF: 6% BTBPE: <MDL, DF: 22% DBDPE: 47 ng/g/lw, DF: 88% TPrP: 2.5 ng/g/lw, DF: 69% TBP: 127 ng/g/lw, DF: 25% TCEP: 15 ng/g/lw, DF: 64% TCPP: 237 ng/g/lw, DF: 61% TDCPP: 12 ng/g/lw, DF: 60% TBEP: 89 ng/g/lw, DF: 71% TPP: 22 ng/g/lw, DF: 81% EHDP: 15 ng/g/lw, DF: 3%
Hu (2021) [42]	BDE-28, BDE-47, BDE-99, BDE-100, BDE-183	China	Cohort	Rural adult residents along Yangtze River (n = 329)	Serum and urine	High relative concentrations; DFs not included BDE-28: 1.49 ng/g lw BDE-47: 0.96 ng/g lw BDE-99: 1.16 ng/g lw BDE-100: 2.04 ng/g lw BDE-183: 2.15 ng/g lw
Percy (2021) [21]	BDCIPP, DPHP	USA	Cohort	Pregnant women and their newborns (n = 298)	Urinary and cord serum	Not specified DPHP: 1.74 µg/g creatinine (16 weeks), 1.73 µg/g creatinine (26 weeks), 2.10 µg/g creatinine (delivery), DF: 95%; BDCIPP: 0.75 µg/g creatinine (16 weeks), 0.74 µg/g creatinine (26 weeks), 0.81 µg/g creatinine (delivery), DF: 100%
Yang (2021) [43]	BDE-47, BDE-153	China	Cross-sectional	Patients with abnormal thyroid hormone levels (n = 40)	Serum	Not specified BDE-47: 0.02 ng/mL, DF: 100% BDE-153: 0.01 ng/mL, DF: 89%
Yao (2021) [24]	DBP, DPHP	China	Case–control	Pregnant women from an urban region and their newborns (n = 360)	Urinary and serum	High DPHP compared to DBP DBP: 0.19 μg/g creatinine, DF: 57.2% DPHP: 0.66 μg/g creatinine, DF: 100%
Zhang (2021) [28]	BDE-209, ∑PBDEs = −28, −47, −99, −100, −153, −154, −183, −209 BDE-28, BDE-47, BDE-183, BDE-209, ∑PBDEs = 28, −47, −99, −100, −153, −154, −183	China	Case–control	Thyroid cancer patients from Anhui province (n = 616)	Serum	High BDE-209 compared to other BDEs BDE-28: 2.54 ng/g lw, DF: 79% BDE-47: 2.009 ng/g lw, DF: 100% BDE-99: 1.16 ng/g lw, DF: 98% BDE-100: 2.04 ng/g lw, DF: 83% BDE-153: 1.20 ng/g lw, DF: 89% BDE-154: 1.73 ng/g lw, DF: 65% BDE-183: 2.15 ng/g lw, DF: 82% BDE-209: 47.91 ng/g lw, DF: 100%
Zhao (2021) [44]	BDE-47, BDE-99, BDE-100, BDE-209	China	Case–control	Residents of a well-known FR production region (n = 172)	Serum	High BDE-209 compared to others BDE-47: 0.607 ng/g lw, DF: 100% BDE-99: 0.600 ng/g lw, DF: 100% BDE-100: 0.333 ng/g lw, DF: 100%
Zhao (2021) [45]	PBEB, DBDPE	China	Cross-sectional	Residents of a BFR-producing region (n = 172)	Serum	PBEB: 0.134 ng/g lipid weight, DF: 94% DBDPE: 32.5 ng/g lipid weight, DF: 98%
Gravel (2020) [46]	tb-DPHP, BDE-209, BDE-47, BDCIPP, BDE-153	Canada	Cross-sectional	Electronic waste recycling workers (n = 100)	Plasma and urine	10× higher BDE-209 concentration in e-waste site workers than the control group tb-DPHP: 0.032 ng/mL, DF: 60% BDE-209: 32 ng/g lipids, DF: 89% BDE-47: 12 ng/g lipids, DF: 42% BDCIPP: 1.3 ng/mL, DF: 50% BDE-153: 8.0 ng/g lipids, DF: 44%
Huang (2020) [47]	BDE-28	USA	Case–control	US military personnel (n = 148)	Serum samples during active duty	Higher concentrations noted to be correlated with a greater risk of PTC BDE-28: < MDL, DF: 33.8%
Kassotis (2020) [48]	TCEP, TDCIPP, 4-tBPDPP, B4tBPPP, T4tBPP, DiNP, TOTM, BDE-100	USA	Case–control	Adults in central North Carolina (n = 72)	Chemical mixtures isolated from personal silicone wristband samplers	Higher concentrations of TDCIPP compared to others TCEP: 22.4 ng/g, DF: 68.9% TDCIPP: 359.6 ng/g, DF: 100% 4-tBPDPP: 59.2 ng/g, DF: 98.6% B4tBPPP: 23.4 ng/g, DF: 91.9% T4tBPP: 3.2 ng/g, DF: 79.7% DiNP: 62,942.5 ng/g, DF: 100% TOTM: 480.3 ng/g, DF: 100% BDE-100: 12.6 ng/g, DF: 79.2%
Li (2020) [49]	BDE-47, BDE-99, BDE-154, BDE-100, BDE-196, BDE-197, BDE-203, BDE-207	Germany	Cohort	Women from the LUPE cohort (n = 99)	Breast milk	Not specified BDE-47: 204 pg/g lw, DF: 100; BDE-99: 62.5 pg/g lw, DF: 99 BDE-154: 8.25 pg/g lw, DF: 83 BDE-100: 54.3 pg/g lw, DF: 98 BDE-196: 16.8 pg/g lw, DF: 90 BDE-197: 73.1 pg/g lw, DF: 100 BDE-203: 16.7 pg/g lw, DF: 92 BDE-207: 56.3 pg/g lw, DF: 99%
Zhao (2020) [50]	DBDPE	China	Case–control	DBDPE manufacturing workers (n = 104)	Hair and nail and serum	High relative to other flame retardants in the region DBDPE: 40.5 ng/g lw, DF: 98%
Chen (2019) [51]	DBDPE	China	Case–control	Adults in a DBDPE manufacturing area (n = 302)	Serum	High DBDPE in manufacturing workers compared to non-working exposed residents DBDPE for occupationally exposed workers: 4170 ng/g lw, DF: 100% DBDPE for residents in manufacturing contaminated areas: 33.4 ng/g lw, DF: 100%
Cowell (2019) [52]	BDE-47	USA	Cohort	Pregnant women in NYC and their children (n = 158)	Plasma and serum	Significant variation in concentration detected BDE-47: 75.0 ng/g lw, DF: 94.4%
Curtis (2019) [53]	PBBs	USA	Cohort	Children exposed to PBEs from the Michigan PBB registry	Serum	High PBB concentrations demonstrated effects relative to low PBB concentrations Before puberty complete (age ≤ 16): 0.22 ppb, DF > 92% After puberty complete (age > 16): 0.72 ppb, DF > 92%
Deziel (2019) [27]	BDE-209	USA	Case–control	Thyroid cancer population in Connecticut (n = 500)	Serum	High exposure levels correlated with a decreased risk of PTC BDE-209: 1.47 ng/g lw, DF: 84%
Guo (2019) [54]	BDE-47, BDE-153, BDE-183, BDE-204, BDE-207 ∑PBDEs	China	Case–control	Residents of an e-waste region (n = 112)	Serum	Higher concentrations at e-waste dismantling site compared to control BDE-47: 6.36 ng/g lw, DF: 99% BDE-153: 3.04 ng/g lw, DF: 100% BDE-183: <MDL, DF: 28% BDE-209: 1.47 ng/g lw, DF: 84%
Wang (2019) [55]	BDE-99, BDE-100, BDE-197, BDE-208, HBB, BDE-47, PBEB, TBE, BEHTBP	USA	Cohort	Residents in rural Central Appalachia (n = 101)	Serum and silicone wristband	High-exposure group compared to low-exposure group demonstrated greater thyroid disruption BDE-99: 11 ng/g lw, DF: 99% BDE-100: 2.2 ng/g lw, DF: 97% BDE-197: 0.065 ng/g lw, DF: 92% BDE-208: 0.14 ng/g lw, DF: 100% HBB: 0.032 ng/g lw, DF: 87% BDE-47: 14 ng/g lw, DF: 98% PBEB: 0.049 ng/g lw, DF: 96% TBE: 0.65 ng/g lw, DF: 96% BEHTBP: 31 ng/g lw, DF: 100%
Albert (2018) [56]	BDE-47	Canada	Cohort	Healthy young men (n = 47)	Serum	Not specified BDE-47: 12.7 ng/g lw, DF: 27.2%
Byrne (2018) [29]	BDE-28/33, BDE-47, BDE-100, BDE-153	USA	Case–control	Remote Alaska Native population (n = 85)	Serum	Not specified BDE-28/33: 3.22 pg/mL ww (wet weight), DF: 100% BDE-47: 46.57 pg/mL ww, DF: 97% BDE-99: 9.19 pg/mL ww, DF: 89% BDE-100: 9.96 pg/mL ww, DF: 97% BDE-153: 59.64 pg/mL ww, DF: 100% BDE-209: 18.39 pg/mL ww, DF: 97%
Chen (2018) [57]	BDE-209	China	Cross-sectional	Occupational workers from a deca-BDE manufacturing plant (n = 72)	Serum and urine	High levels of BDE-209 relative to other studies and the general population BDE-209: 3420 ng/g lw, DF: 100%
Guo (2018) [30]	BDE-47, BDE-183, BDE-209	China	Case–control	Fifth graders from South China (n = 174)	Serum	High BDE-209 concentration relative to other flame retardants in students living near a petrochemical complex BDE-47: 4.4 ng/g lw, DF: 100% BDE-183: 2.4 ng/g lw, DF: 93% BDE-209: 95 ng/g lw, DF: 98%
Li (2018) [58]	BDE-99, BDE-100, ∑PBDEs (−47, −99, −100)	Denmark	Cohort	Mothers of boys with and without cryptorchidism (n = 58)	Placenta	High exposure of ∑PBDEs relative to the general population BDE-99: <LOD (below the limit of detection), DF: 20.8% BDE-100: <LOD (below the limit of detection), DF: 12.5% ∑PBDEs: 3710 ng/g lw, DF: 100%
Vuong (2018) [59]	∑PBDEs (BDE-28, BDE-47, BDE-99, BDE-100, BDE-153)	USA	Cohort	Mother–child pairs (n = 162)	Serum	Not specified BDE-47: 72.2 ng/g lw, DF: 100% BDE-99: 18.7 ng/g lw, DF: 100% BDE-100: 11.7 ng/g lw, DF: 100% ∑PBDEs: 105.8 ng/g lw, DF: 100%
Ding (2017) [19]	BDE-99, ∑PBDEs (−47, −99, −100, −153)	China	Cohort	Pregnant women in rural northern China (n = 107)	Cord blood	10× increase in BDE-99 and ∑PBDEs each associated with an approximately 5% increase in tT4 levels BDE-99: 8.27 ng/g lw, DF: 97.2% ∑PBDEs: Not explicitly provided
Hoffman (2017) [2]	BDE-209, TCEP	USA	Case–control	Patients with papillary thyroid cancer at Duke University Hospital (n = 140)	Serum	Higher levels of BDE-209 are associated with smaller and less aggressive PTC, but a higher level of TCEP is associated with larger more aggressive PTC BDE-209: 95 ng/g lw, DF: 98% TCEP: Not explicitly stated
Liu (2017) [31]	OH-BDE-49, OH-BDE-47, ∑OH-PBDEs (−47, −49, −42)	China	Case–control	Thyroid cancer patients (n = 33)	Serum	Not specified OH-BDE-49: 0.007 ng/g lw, DF: 57.6% OH-BDE-47: 0.01 ng/g lw, DF: 72.7% ΣOH-PBDEs (−47, −49, −42): 0.06 ng/g lw, DF: 100%
Preston (2017) [60]	DPHP	USA	Cohort	Office workers from the Boston area (n = 51)	Serum and urine	Variable DPHP concentrations relative to other studies DPHP: 2.65 ng/mL (high concentration threshold), DF: 95%
Zheng (2017) [61]	BDE-47, BDE-66, BDE-85	China	Cohort	Occupational e-waste recycling workers (n = 79)	Serum	Variable concentrations compared to other studies depending on location and population BDE-47: 4.4 ng/g lw, DF: 100% BDE-66: 0.77 ng/g lw, DF: 80% BDE-85: 1.42 ng/g lw, DF: 60.4%
Zheng (2017) [62]	BDE-153, BDE-7, BDE-99	China	Cohort	Pregnant women (n = 72)	Women and cord serum	Concentrations in this study generally lower than in North America but higher than in Asia and Europe BDE-153: 0.43 ng/g lw, DF: 87.9% BDE-7: Not detected BDE-99: 0.31 ng/g lw, DF: 84.8%
Jacobson (2016) [63]	PBDE-47, PBDE-99, PBDE-100, PBDE-153, ∑PBDEs	USA	Cohort	Pediatric anesthesia patients, ages 1–5 in Atlanta (n = 80)	Serum	Similarly high concentrations of PBDE compared to studies within the US PBDE-47: 0.15 ng/g lw, DF: 100% PBDE-99: 0.04 ng/g lw, DF: 100% PBDE-100: 0.02 ng/g lw, DF: 83.8% PBDE-153: 0.02 ng/g lw, DF: 63.8% ∑PBDEs: 0.25 ng/g lw, DF: 100%
Leonetti (2016) [64]	BDE-47, BDE-99, BDE-209, 2,4,6-TBP, ∑BFR	USA	Cohort	Women who delivered term infants (n = 95)	Placenta	Higher relative concentrations of placental 2,4,6-TBP compared to PBDEs BDE-47: 5.45 ng/g lw, DF: 91.2% BDE-99: 2.02 ng/g lw, DF: 68.6% BDE-209: 3.08 ng/g lw, DF: 52.9% 2,4,6-TBP: 15.8 ng/g lw, DF: 100% ΣBFR: 39.1 ng/g lw, DF: Not specified
Lignell (2016) [65]	BDE-153	Sweden	Cross-sectional	Randomly selected mothers from Uppsala County (n = 126)	Serum and breast milk	10× lower PBDE body burden in Sweden women in this study compared to the US BDE-153: 0.48 ng/g lw, DF: Not specified
Aschebrook-Kilfoy (2015) [66]	BDE-47, BDE-99, BDE-100, BDE-153	USA	Case–control	Nested CC in the prostate, lung, colorectal, and ovarian cancer screening trial (n = 311)	Serum	High BDE-47 burden compared to other PBDEs (70.5% of all included PBDEs) BDE-47: 12.9 ng/g lw, DF: 93.9% BDE-99: 2.8 ng/g lw, DF: 58.5% BDE-100: 1.7 ng/g lw, DF: 65.6% BDE-153: 1.6 ng/g lw, DF: 65.3%
Kim (2015) [20]	BDE-47, BDE-99	Korea	Cross-sectional	Newborn infant population (n = 104)	Cord serum and bloodspot	Lower concentrations of PBDEs in this population of Korean infants compared to US infants but higher than European infants BDE-47: 3.0 ng/g lw, DF: 74.0% BDE-99: 3.0 ng/g lw, DF: 64.4%
Makey (2016) [67]	BDE-47, BDE-153	USA	Cohort	Healthy adult office workers in Boston (n = 52)	Serum	Higher concentration than Asian and European studies BDE-47: 9.5 ng/g lw, DF: 100% BDE-153: 6.4 ng/g lw, DF: 100%
Vuong (2015) [23]	BDE-28, BDE-47	USA	Cohort	Pregnant women from the HOME study (n = 389)	Serum and cord blood	Higher concentrations of PBDEs by 10–100× compared to European and Japanese studies BDE-28: 1.0 ng/g lw, DF: 80.0% BDE-47: 19.1 ng/g lw, DF: 100%
Xu (2015) [68]	∑PBDEs	China	Case–control	Residents of an e-waste dismantling area in Zhejiang (n = 55)	Serum	Higher concentration of population near e-waste dismantling sites compared to control ΣPBDEs: 139.32 ng/g lw, DF: Not specified
Bloom (2014) [69]	∑BDEs	USA	Cross-sectional	Upper Hudson River communities (n = 253)	Serum	Average levels of BDEs in this study relatively lower in a similar US population ΣBDEs: 0.42 µg/L serum (approximately 42 ng/g lw), DF: 100%
Huang (2014) [70]	BDE-17, BDE-28, BDE-47, BDE-99, BDE-153, BDE-183, BDE-209	China	Cohort	Volunteers from northern China (n = 124)	Serum	Median BDE concentrations comparable to northern China, but generally lower than southern China BDE-17: below limit of quantification (bLOQ), DF: not specified BDE-28: 0.25 ng/g lw, DF: Not specified BDE-47: 0.21 ng/g lw, DF: Not specified BDE-99: 0.20 ng/g lw, DF: Not specified BDE-153: 0.62 ng/g lw, DF: 95% BDE-183: 0.22 ng/g lw, DF: Not specified BDE-209: 5.02 ng/g lw, DF: Not specified
Abdelouahab (2013) [16]	PBDE-47, PBDE-99, ∑PBDEs	Canada	Cohort	Pregnant women without thyroid disease (n = 260)	Serum	Lower levels of PBDEs compared to the general US population PBDE-47: 21.47 ng/g lw, DF: 100% PBDE-99: 2.32 ng/g lw, DF: 96% ΣPBDEs: 30.92 ng/g lw, DF: 100%
Johnson (2013) [71]	PentaBDE = ∑BDE-47, BDE-99, BDE-100 OctaBDE = ∑BDE-183 and BDE-201	USA	Cohort	Men recruited from Massachusetts General Hospital (n = 62)	Serum	Higher concentrations of pentaBDE than in European countries PentaBDE (∑BDE-47, BDE-99, BDE-100): 1049 ng/g dust, DF: 100% OctaBDE (∑BDE-183 and BDE-201): 30.5 ng/g dust, DF: 100%
Kim (2013) [72]	BDE-28, BDE-47, ∑PBDEs	Korea	Cohort	Pregnant women in Korea (n = 138)	Serum	Lower PBDE concentration in this population compared to a similar North American population BDE-28: 0.32 ng/g lw, DF: 68% BDE-47: 9.5 ng/g lw, DF: 100% ΣPBDEs: 18.7 ng/g lw, DF: 100%
Kim (2012) [73]	BDE-53, BDE-49, BDE-153, BDE-154, BDE-196, BDE-197	South Korea	Case–control	Children with congenital hypothyroidism and their mothers (n = 76)	Serum	Lower PBDE concentrations compared to North American studies, but higher than values reported in the Netherlands BDE-53: 1.2 ng/g lw, DF: 38% BDE-49: 0.5 ng/g lw, DF: 50% BDE-153: 6.4 ng/g lw, DF: 100% BDE-154: 0.6 ng/g lw, DF: 83% BDE-196: 0.9 ng/g lw, DF: 25% BDE-197: 0.7 ng/g lw, DF: 21%
Leijs (2012) [74]	BDE-99	Netherlands	Cohort	14–19-year-old children from Amsterdam/Zaandam region (n = 33)	Serum	The concentration of BDE in the Netherlands was relatively high compared to other European countries, but still low compared to the US BDE-99: 1.6 ng/g lw, DF: 88%
Chevrier (2011) [18]	BDE-17, BDE-28, BDE-47, BDE-66, BDE-85, BDE-99, BDE-100, BDE-153, BDE-154, BDE-183	USA	Cross-sectional	Pregnant women in California Salinas Valley and their children (n = 289)	Serum	BDE-47 dominant congener in this study, consistent with prior studies Variable PBDE levels compared to other countries BDE-17: <LOD, DF: 1% BDE-28: 0.6 ng/g lw, DF: 57.6% BDE-47: 15.2 ng/g lw, DF: 99.7% BDE-66: <LOD, DF: 18.8% BDE-85: 0.3 ng/g lw, DF: 51.0% BDE-99: 3.8 ng/g lw, DF: 99.0% BDE-100: 2.6 ng/g lw, DF: 97.6% BDE-153: 2.2 ng/g lw, DF: 96.9% BDE-154: <LOD, DF: 49.0% BDE-183: <LOD, DF: 30.2%
Eggesbø (2011) [7]	BDE-28, 47, 99, 153, 154, 209 and HBCD	Norway	Cohort	Women in the Norwegian human milk study (n = 239)	Serum and breast milk	Lower flame retardant concentrations in Norway than in comparable US or Canadian populations BDE-28: 0.25 ng/g lw, DF: Not specified BDE-47: 0.21 ng/g lw, DF: 92% BDE-99: 0.20 ng/g lw, DF: 91% BDE-153: 0.62 ng/g lw, DF: 95% BDE-154: 0.08 ng/g lw, DF: 2% BDE-209: 5.02 ng/g lw, DF: 121 HBCD: 1.24 ng/g lw, DF: 66%
Lin (2011) [75]	BDE-99, BDE-154, BDE-183, ∑PBDEs	Taiwan	Cohort	Mothers and their nursing infants with PBDE exposure (n = 54)	Serum and cord blood	Cord blood from Taiwanese newborns was significantly lower than reported in American studies BDE-99: 0.724 ng/g lw, DF: 83.3% BDE-154: 0.100 ng/g lw, DF: 90.7% BDE-183: 0.505 ng/g lw, DF: 50.0% ∑PBDEs: 3.49 ng/g lw, DF: N/A
Stapleton (2011) [22]	BDE-47/99/100, BDE-153, 4′OH-BDE-49/6-OH-BDE-47	USA	Case–control	Pregnant women >34 weeks into pregnancy (n = 137)	Serum	Notable temporal decrease in PBDE concentration from prior studies BDE-47: 18.87 ng/g lw, DF: 94.89% BDE-99: 5.50 ng/g lw, DF: 64.23% BDE-100: 4.61 ng/g lw, DF: 89.05% BDE-153: 5.65 ng/g lw, DF: 96.35% 4′-OH-BDE-49: 0.12 ng/g lw, DF: 71.93% 6-OH-BDE-47: 0.19 ng/g lw, DF: 66.67%
Zota (2011) [26]	BDE-85, BDE-207	USA	Cohort	Second-trimester pregnant women (n = 25)	Serum	The BDE concentrations were the highest reported concentrations in pregnant women at the time BDE-85: 0.82 ng/g lw, DF: 72% BDE-207: 1.54 ng/g lw, DF: 52%
Chevrier (2010) [17]	∑PBDEs (BDE-28, BDE-47, BDE-99, BDE-100, BDE-153)	USA	Case–control	Pregnant women from Monterey County, CA (n = 270)	Serum	The BDE concentrations were the highest reported concentrations at the time BDE-28: 0.5 ng/g lw, DF: 52.2% BDE-47: 15.0 ng/g lw, DF: 99.6% BDE-99: 4.0 ng/g lw, DF: 99.6% BDE-100: 2.4 ng/g lw, DF: 98.5% BDE-153: 2.1 ng/g lw, DF: 98.5% ∑PBDEs: 25.2 ng/g lw, DF: 100%
Wang (2010) [76]	BDE-126, BDE-205, PBB-103	China	Case–control	People exposed to an e-waste site (n = 325)	Serum	Relative higher levels of BDEs by e-waste dismantling regions compared to controls BDE-126: 0.19 ng/mL plasma, DF: Not specified BDE-205: 0.03 ng/mL plasma, DF: Not specified PBB-103: 0.67 ng/mL plasma, DF: Not specified
Dallaire (2009) [33]	BDE-47, BDE-153	Canada	Cross-sectional	Inuit adults (n = 623)	Serum	Average concentrations significantly lower in this population than the average US population BDE-47: 2.16 ng/g lw, DF: 57.3% BDE-153: 2.05 ng/g lw, DF: 73.8%
Turyk (2008) [37]	BDE-47, ∑PBDEs	USA	Cohort	Adult male sport fish consumers (n = 354)	Serum and urine	Not specified BDE-47: 3.8 ng/g lw, DF: 98% ∑PBDEs: 38 ng/g lw, DF: 100%
Herbstman (2008) [34]	BDE-47, BDE-100, BDE-153	USA	Cohort	Infants delivered at Johns Hopkins Hospital (n = 297)	Serum	Relative concentrations consistent with average reported US BDE concentrations BDE-47: 13.8 ng/g lw, DF: 90.7% BDE-100: 2.3 ng/g lw, DF: 35.6% BDE-153: 2.6 ng/g lw, DF: 39.8%
Julander (2005) [35]	BDE-28, BDE-99, BDE-100, BDE-154, BDE-183	Sweden	Cohort	Personnel working with electronic dismantling (n = 19)	Serum	BDE-154 and BDE-183 notably higher than the general non-e-waste occupational Swedish population, while BDE-99 and BDE-100 were lower; DF was not reported BDE-28: 0.25 pmol/g lw BDE-99: 0.78 pmol/g lw BDE-100: 0.44 pmol/g lw BDE-154: 0.19 pmol/g lw BDE-183: 0.83 pmol/g lw
Bahn (1980) [32]	PBB	USA	Cohort	Workers from a PBB manufacturing plant (n = 86)	Serum	Higher relative concentrations in occupations including electronic recycling and steel workers compared to the general population PBB: 1.5 ng/mL serum, DF: 87%

Abbreviations: DF = distribution frequency, lw = lipid weight.

**Table 2 biomedicines-12-01365-t002:** Flame retardant and TSH association.

Author (Year), Ref.	Flame Retardant Type	Country	Study Design	Investigated Population (n)	Measurement of Exposure	Association with TSH
Liu (2023) [39]	TDCIPP, TCIPP, TEP, TCEP	China	Cohort	Females of childbearing age (n = 319)	Serum	**TDCIPP (β = 0.12, *p* < 0.05, 95% CI: 0.02, 0.22)** **TEP (β = 0.25, *p* < 0.01, 95% CI: 0.08, 0.41)—Group B** **TEP (β = 0.27, 95% CI: 0.01, 0.54, *p* < 0.05)—Group D** **TCIPP (β = 0.09, *p* < 0.05, 95% CI: 0.006, 0.17)**
Babichuk (2023) [38]	PBB-153, PBDE-28, PBDE-47, PBDE-99, PBDE-100, PBDE-153	Canada	Cohort	Two rural coastal populations (n = 80)	Serum	PBB-153 (β = −0.089, *p* = 0.755, 95% CI: −0.695, 2.375) PBDE-28 (β = −0.410, *p* = 0.183, 95% CI: −1.867, 0.364) PBDE-47 (β = 0.006, *p* = 0.989, 95% CI: −0.111, 0.113) PBDE-99 (β = 0.256, *p* = 0.364, 95% CI: −0.201, 0.540) PBDE-100 (β = 0.286, *p* = 0.201, 95% CI: −0.032, 0.151) PBDE-153 (β = −0.214, *p* = 0.225, 95% CI: −0.029, 0.007)
Liu (2022) [41]	PBT, DBDPE, TCEP, TPP, EHDP	China	Case–control	Patients with or without thyroid cancer (n = 481)	Serum	PBT (β = 1.63, *p* > 0.05, 95% CI: −9.97, 14.68) DBDPE (β = −3.63, *p* > 0.05, 95% CI: −10.51, 3.87) TCEP (β = 13.12, *p* > 0.05, 95% CI: −0.60, 28.66) TPP (β = −6.48, *p* > 0.05, 95% CI: −17.76, 6.24) EHDP (β = −5.34, *p* > 0.05, 95% CI: −15.28, 5.77)
Yao (2021) [24]	DBP, DPHP	China	Case–control	Pregnant women from an urban region and their newborns (n = 360)	Urinary and serum	**DBP (β = 0.277, 95% CI: 0.104, 0.449)—Newborns** **DPHP (β = 0.061, 95% CI: 0.027, 0.095)—Mothers**
Gravel (2020) [46]	tb-DPHP, BDE-209, BDE-47, BDCIPP, BDE-153	Canada	Cross-sectional	Electronic waste recycling workers (n = 100)	Plasma and urine	tb-DPHP: β = −0.362, *p* > 0.05, 95% CI: −1.489, 0.764 BDE-209: β = −0.021, *p* > 0.05, 95% CI: −0.158, 0.115 BDE-47: β = 0.009, *p* > 0.05, 95% CI: −0.017, 0.035 BDCIPP: β = 0.163, *p* > 0.05, 95% CI: −0.541, 0.866 BDE-153: β = 0.107, *p* > 0.05, 95% CI: −0.100, 0.313
Guo (2019) [54]	BDE-47, BDE-100, TBECH	China	Case–control	Residents of an e-waste region (n = 112)	Serum	**BDE-47 (β = −0.31, 95% CI: −0.48, −0.14)** **BDE-100 (β = −0.15, 95% CI: −0.30, −0.0044)** **TBECH (β = −0.096 95% CI: −0.19, −0.0061)**
Cowell (2019) [52]	BDE-47	USA	Cohort	Pregnant women in NYC and their children (n = 158)	Plasma and serum	**BDE-47 (β = −0.09, 95% CI: −0.16, −0.02)**
Chen (2019) [51]	DBDPE	China	Case–control	Adults in a DBDPE manufacturing area (n = 302)	Serum	DBDPE (β = 0.009, 95% CI: −0.015, 0.032)
Vuong (2018) [59]	∑PBDEs (BDE-28, BDE-47, BDE-99, BDE-100, BDE-153)	USA	Cohort	Mother–child pairs (n = 162)	Serum	**∑PBDEs (β = −0.32, 95% CI −0.53, −0.12)**
Byrne (2018) [29]	∑PBDEs (−28, −33), BDE-47, BDE-100	USA	Case–control	Remote Alaska Native population (n = 85)	Serum	**∑PBDEs (β = 0.41, *p* < 0.001, 95% CI:0.19, 0.63)** **BDE-47 (β = 3.87, *p* < 0.005, 95% CI: 1.21, 6.57)** **BDE-100 (β = 0.89, *p* = 0.01, 95% CI: 0.18, 1.61)**
Albert (2018) [56]	BDE-47	Canada	Cohort	Healthy young men (n = 47)	Serum	**BDE-47 (β = −0.17, 95% CI: −31.5, 0.0, *p* = 0.05)**
Liu (2017) [31]	OH-BDE-49, ∑OH-PBDEs (PBDE-47, PBDE-49, PBDE-42)	China	Case–control	Thyroid cancer patients (n = 33)	Serum	**OH-BDE-49 (β = 0.33, 95% CI: 0.04, 0.42)** **∑OH-BDEs (β = 0.36, 95% CI: 0.07, 0.64)**
Jacobson (2016) [63]	PBDE-47, PBDE-99, PBDE-100, PBDE-153, ∑PBDEs	USA	Cohort	Pediatric anesthesia patients, ages 1–5 in Atlanta (n = 80)	Serum	**PBDE-47 (β = 0.20, 95% CI: 0.02, 0.37)** **PBDE-99 (β = 0.20, 95% CI: 0.04, 0.36)** **PBDE-100 (β = 0.10, 95% CI: 0.02, 0.18)** PBDE-153 (β = 0.05, 95% CI: −0.11, 0.22) **∑PBDEs (β = 0.20, 95% CI: 0.02, 0.39)**
Kim (2015) [20]	BDE-47, BDE-99	Korea	Cross-sectional	Newborn infant population (n = 104)	Cord serum and bloodspot	**BDE-47 (β = 0.327, *p* < 0.05, 95% CI: 0.03, 0.62)—bloodspot** **BDE-99 (β = 0.211, *p* < 0.05, 95% CI: 0.00, 0.42)—cord**
Huang (2014) [70]	BDE-17, BDE-28, BDE-47, BDE-99, BDE-183	China	Cohort	Volunteers from northern China (n = 124)	Serum	**BDE-17 (r = 0.459, *p* < 0.01)** **BDE-28 (r = 0.308, *p* < 0.01)** **BDE-47 (r = 0.211, *p* < 0.05)** **BDE-99 (r = −0.252, *p* < 0.01)** **BDE-183 (r = 0.280, *p* < 0.01)**
Johnson (2013) [71]	OctaBDE = ∑BDE-183 and BDE-201	USA	Cohort	Men recruited from Massachusetts General Hospital (n = 62)	Serum	OctaBDE (β = 21.2, *p* = 0.05, 95% CI: 0.8, 45.8)
Kim (2012) [73]	BDE-154, BDE-153, BDE-197, BDE-196	South Korea	Case–control	Children with congenital hypothyroidism and their mothers (n = 76)	Serum	**BDE-154 (r = 0.641, *p* < 0.05)** **BDE-153 (r = 0.591, *p* < 0.05)** **BDE-197 (r = 0.818, *p* < 0.01)** **BDE-196 (r = 0.794, *p* < 0.01)**
Zota (2011) [26]	BDE-85, BDE-207	USA	Cohort	Second-trimester pregnant women (n = 25)	Serum	**BDE-85 (β = 0.33, *p* < 0.05, 95% CI: 0.02, 0.64)** **BDE-207 (β = −0.72, *p* < 0.01, 95% CI: −1.10, −0.34)**
Eggesbø (2011) [7]	∑PBDEs and HBCD	Norway	Cohort	Women in the Norwegian human milk study (n = 239),	Breast milk, serum	∑PBDEs (β = 0.00, 95% CI: −0.01, 0.02) HBCD (β = 0.00, 95% CI: −0.02, 0.02)
Chevrier (2011) [18]	BDE-17, BDE-28, BDE-47, BDE-66, BDE-85, BDE-99, BDE-100, BDE-153, BDE-154, BDE-183	USA	Cross-sectional	Pregnant women in California Salinas Valley and their children (n = 289)	Serum	∑PBDEs (β = 0.00, 95% CI: −0.06, 0.06)
Chevrier (2010) [17]	∑PBDEs (BDE-28, BDE-47, BDE-99, BDE-100, BDE-153)	USA	Case–control	Pregnant women from Monterey County, CA (n = 270)	Serum	**∑PBDEs (β = −0.08, *p* < 0.05, 95% CI: −0.14, −0.01)** **BDE-28 (β = −0.05, *p* < 0.05, 95% CI: −0.10, −0.00)** **BDE-47 (β = −0.07, *p* < 0.05, 95% CI: −0.13, −0.01)** **BDE-100 (β = −0.09, *p* < 0.01, 95% CI: −0.15, −0.02)** **BDE-99 (β = −0.07, *p* < 0.05, 95% CI: −0.13, −0.00)** **BDE-153 (β = −0.08, *p* < 0.05, 95% CI: −0.15, −0.01)**
Turyk (2008) [37]	BDE-47	USA	Cohort	Adult male sport fish consumers (n = 354)	Urinary and serum	**BDE-47 (r = −0.14, *p* = 0.02)**
Herbstman (2008) [34]	BDE-47, BDE-100	USA	Cohort	Infants delivered at Johns Hopkins Hospital (n = 297)	Serum	**BDE-47 (β = 0.39, 95% CI: 0.19, 0.78)** **BDE-100 (β = 0.36, 95% CI: 0.16, 0.82)**
Julander (2005) [35]	BDE-99, BDE-154	Sweden	Cohort	Personnel working with electronic dismantling (n = 19)	Serum	**BDE-99 (r = 0.79 *p* = 0.036)** **BDE-154 (r = 0.80 *p* = 0.031)**
Bahn (1980) [32]	PBB	USA	Cohort	Workers from a PBB manufacturing plant (n = 86)	Serum	**PBB (*p* = 0.006)**

Bolded studies indicate statistically significant findings.

**Table 3 biomedicines-12-01365-t003:** Flame retardant and tT3.

Author (Year), Ref.	Flame Retardant Type	Country	Study Design	Investigated Population (n)	Measurement of Exposure	Association with T3
Liu (2022) [41]	PBT, DBDPE, TCEP, TPP, EHDP	China	Case–control	Patients with or without thyroid cancer (n = 481)	Serum	**PBT (β = −3.41, *p* < 0.05, 95% CI: −6.42, −0.30**) DBDPE (β = −0.43, *p* > 0.05, 95% CI: −2.37, 1.56) TCEP (β = −3.00, *p* > 0.05, 95% CI: −6.26, 0.3**8**) **TPP (β = 6.72, *p* < 0.01, 95% CI: 3.25, 10.30)** EHDP (β = 3.67, *p* < 0.05, 95% CI: 0.71, 6.74)
Zhao (2021) [45]	PBEB, DBDPE	China	Cross-sectional	Residents of a BFR-producing region (n = 172)	Serum	**PBEB (β = 0.031, 95% CI 0.001, 0.060)** **DBDPE (β = −0.037, 95% CI −0.070, −0.003)**
Zhao (2021) [44]	BDE-47, BDE-99	China	Case–control	Residents of a well-known FR production region (n = 172)	Serum	**BDE-47 (β = 0.039, 95% CI: 0.001, 0.078)** **BDE-99 (β = 0.032, 95% CI: 0.005, 0.060)**
Percy (2021) [21]	DPHP	USA	Cohort	Pregnant women and their newborns (n = 298)	Urinary and cord serum	DPHP Q2 (β = 1.06 95% CI: 0.86, 1.30) DPHP Q3 (β = 1.23, 95% CI: 1.00, 1.52) **DPHP Q4 (β = 1.25, 95% CI: 1.01, 1.54)**
Yang (2021) [43]	BDE-153	China	Cross-sectional	Patients with abnormal thyroid hormone levels (n = 40)	Serum	**BDE-153 (β = −0.97, *p* < 0.05, 95% CI: −1.74, −0.20)**
Zhao (2020) [50]	DBDPE	China	Case–control	DBDPE manufacturing workers (n = 104)	Hair and nail, serum	**DBDPE (r = 0.214 *p* = 0.025)**
Li (2020) [49]	BDE-47, BDE-99, BDE-100, BDE-197, BDE-203, BDE-207	Germany	Cohort	Women from the LUPE cohort (n = 99)	Breast milk	**BDE-47 (β = −0.12, 95% CI: −0.22, −0.02)** **BDE-99 (β = −0.10, 95% CI: −0.21, −0.002)** **BDE-100 (β = −0.12, 95% CI: −0.22, −0.02)** **BDE-197 (β = −0.11, 95% CI: −0.21, −0.01)** **BDE-203 (β = −0.14, 95% CI: −0.24, −0.03)** **BDE-207 (β = −0.11, 95% CI: −0.20, −0.01)**
Gravel (2020) [46]	tb-DPHP, BDE-209, BDE-47, BDCIPP, BDE-153	Canada	Cross-sectional	Electronic waste recycling workers (n = 100)	Plasma and urine	tb-DPHP (β = −0.362, *p* > 0.05, 95% CI: −1.489, 0.764) BDE-209 β = −0.021, *p* > 0.05, 95% CI: −0.158, 0.115) BDE-47 β = 0.009, *p* > 0.05, 95% CI: −0.017, 0.035) BDCIPP β = 0.163, *p* > 0.05, 95% CI: −0.541, 0.866) BDE-153 β = 0.107, *p* > 0.05, 95% CI: −0.100, 0.313)
Guo (2019) [54]	BDE-47, BDE-85, BDE-99, BDE-204, TBECH	China	Case–control	Residents of an e-waste region (n = 112)	Serum	**BDE-47 (β = 0.070, 95% CI: 0.0053, 0.13)** **BDE-85 (β = 0.034, 95% CI: 0.0024, 0.066)** **BDE-99 (β = 0.066, 95% CI: 0.0044, 0.13)** **BDE-204 (β = 0.062, 95% CI: 0.016, 0.11)** **TBECH (β = 0.037, 95% CI: 0.0053, 0.069)**
Chen (2019) [51]	DBDPE	China	Case–control	Adults in a DBDPE manufacturing area (n = 302)	Serum	**DBDPE (β = 0.046, 95% CI: 0.012, 0.081)**
Guo (2018) [30]	BDE-209	China	Case–control	Fifth graders from South China (n = 174)	Serum	**BDE-47 (β = 0.080, *p* < 0.05, 95% CI: 0.011, 0.15)**
Byrne (2018) [29]	BDE-153	USA	Case–control	Remote Alaska Native population (n = 85)	Serum	**BDE-153 (β = −113.14, *p* = 0.05, 95% CI: −225.04, −1.14)**
Zheng (2017) [61]	BDE-47, BDE-66, BDE-85	China	Cohort	Occupational e-waste recycling workers (n = 79)	Serum	**BDE-47 (β = 0.161, *p* = 0.005, 95% CI 0.051, 0.271)** **BDE-66 (β = 0.117, *p* = 0.010, 95% CI 0.029, 0.204)** **BDE-85 (β = 0.172, *p* = 0.004, 95% CI 0.056, 0.288)**
Lignell (2016) [65]	BDE-153	Sweden	Cross-sectional	Randomly selected mothers from Uppsala County (n = 126)	Serum and breast milk	**BDE-153 (β = −0.20 ± 0.08, *p* < 0.05)**
Xu (2015) [68]	∑PBDEs	China	Case–control	Residents of an e-waste dismantling area in Zhejiang (n = 55)	Serum	∑PBDEs (r = 0.13, *p* = 0.342)
Vuong (2015) [23]	BDE-47	USA	Cohort	Pregnant women from the HOME study (n = 389)	Serum and cord serum	**BDE-47 (β = 8.71, *p* < 0.05, 95% CI: 0.42, 16.99)**
Huang (2014) [70]	BDE-17, BDE-28, BDE-47, BDE-99, BDE-153, BDE-183, BDE-209	China	Cohort	Volunteers from northern China (n = 124)	Serum	**BDE-17 (r = −0.444, *p* < 0.01)** **BDE-28 (r = −0.264, *p* < 0.01)** **BDE-47 (r = −0.233, *p* < 0.01)** **BDE-99 (r = 0.324, *p* < 0.01)** **BDE-153 (r = −0.221, *p* < 0.05)** **BDE-183 (r = −0.344, *p* < 0.01)** **BDE-209 (r = 0.254, *p* < 0.05)**
Kim (2013) [72]	∑PBDEs, BDE-47	Korea	Cohort	Pregnant women in Korea (n = 138)	Serum	**∑PBDEs (β = −0.112, *p* < 0.05, 95% CI: −0.170, −0.054)** **BDE-47 (β = −0.042, *p* < 0.05, 95% CI: −0.084, −0.000)**
Johnson (2013) [71]	PentaBDE = ∑BDE-47, BDE-99, BDE-100	USA	Cohort	Men recruited from Massachusetts General Hospital (n = 62)	Serum	**PentaBDE (β = 5.4, *p* = 0.05, 95% CI: 0.0, 10.7)**
Abdelouahab (2013) [16]	PBDE-47, PBDE-99, ∑PBDEs	Canada	Cohort	Pregnant women without thyroid disease (n = 260)	Serum	**PBDE-47 (β = −7.81, 95% CI: −11.37, −4.26)** **PBDE-99 (β = −4.19, 95% CI: −8.26, −0.12)** **∑PBDEs (β = −8.92, 95% CI: −12.63, −5.21)**
Leijs (2012) [74]	BDE-99	Netherlands	Cohort	14–19-year-old children from Amsterdam/Zaandam region (n = 33)	Serum	**BDE-99 (*p* = 0.003)**
Kim (2012) [73]	BDE-154, BDE-53	South Korea	Case–control	Children with congenital hypothyroidism and their mothers (n = 76)	Serum	**BDE-154 (r = −0.577, *p* < 0.05)—Babies** **BDE-153 (r = −0.597, *p* < 0.05)—Mothers**
Lin (2011) [75]	BDE-99, BDE-154, BDE-183	Taiwan	Cohort	Mothers and their nursing infants with PBDE exposure (n = 54)	Serum	**BDE-99 (r = −0.327 *p* = 0.017)** **BDE-154 (r = 0.314 *p* = 0.022)** **BDE-183 (r = 0.271 *p* = 0.049)**
Wang (2010) [76]	PBB-103	China	Case–control	People exposed to an e-waste site (n = 325)	Serum	**PBB-103 (β = −0.11, SE= 0.03 *p* = 0.000)**
Dallaire (2009) [33]	BDE-47	Canada	Cross-sectional	Inuit adults (n = 623)	Serum	**BDE-47 (β = 0.010 *p* < 0.001)**
Julander (2005) [35]	BDE-183	Sweden	Cohort	Personnel working with electronic dismantling (n = 19)	Serum	**BDE-183 (r = 0.93 *p* = 0.003)**

Bolded studies indicate statistically significant findings.

**Table 4 biomedicines-12-01365-t004:** Flame retardant and tT4.

Author (Year), Ref.	Flame Retardant Type	Country	Study Design	Investigated Population (n)	Measurement of Exposure	Association with T4
Trowbridge (2022) [40]	BDCPP	USA	Cross-sectional	Female firefighters and office workers from San Francisco (n = 165)	Urine	**BDCPP (%Δ = −1.95%, 95% CI: −3.57, −0.29)—full group** **BDCPP (%Δ = −2.88%, 95% CI: −5.28, −0.42)—firefighters only**
Liu (2022) [41]	PBT, DBDPE, TCEP, TPP, EHDP	China	Case–control	Patients with or without thyroid cancer (n = 481)	Serum	PBT (β = 0.54, *p* > 0.05, 95% CI: −4.58, 5.92) DBDPE (β = −1.13, *p* > 0.05, 95% CI: −4.25, 2.10) TCEP (β = −1.48, *p* > 0.05, 95% CI: −6.87, 4.21) TPP (β = −2.53, *p* > 0.05, 95% CI: −2.53, 8.88) EHDP (β = −0.30, *p* > 0.05, 95% CI: −4.98, 4.61)
Yang (2021) [43]	BDE-153	China	Cross-sectional	Patients with abnormal thyroid hormone levels (n = 40)	Serum	**BDE-153 (β = 1.11, *p* < 0.05, 95% CI: −0.1, 2.23)**
Li (2020) [49]	BDE-99, BDE-154, BDE-196	Germany	Cohort	Women from the LUPE cohort (n = 99)	Breast milk	**BDE-99 (β = −0.16, 95% CI: −0.28, −0.04)** **BDE-154 (β = −0.14, 95% CI: −0.25, −0.02)** **BDE-196 (β = −0.13, 95% CI: −0.25, −0.003)**
Gravel (2020) [46]	tb-DPHP, BDE-209, BDE-47	Canada	Cross-sectional	Electronic waste recycling workers (n = 100)	Plasma and urine	**tb-DPHP (β = −0.041, 95%, *p* = 0.05 CI: −0.079, −0.001)** **BDE-209 (β = 0.031, *p* = 0.04, 95% CI: 0.0007, 0.061)** BDE-47 (β = 0.05, *p* > 0.05, 95% CI: −0.06, 0.16)
Guo (2019) [54]	BDE-153, BDE-183, ∑PBDEs	China	Case–control	Residents of an e-waste region (n = 112)	Serum	**BDE-153 (β = −3.5, 95% CI: −6.8, −0.12)** **BDE-183 (β = −3.6, 95% CI: −6.4, −0.74)** **∑PBDEs (β = −4.6, 95% CI: −9.1, −0.043)**
Chen (2019) [51]	DBDPE	China	Case–control	Adults in a DBDPE manufacturing area (n = 302)	Serum	**DBDPE (β = 4.73, 95% CI: 2.75, 6.71)**
Li (2018) [58]	BDE-99, BDE-100, ∑PBDEs (−47, −99, −100)	Denmark	Cohort	Mothers of boys with and without cryptoorchidism (n = 58)	Placenta	**BDE-99 (β = −20.2, 95% CI: −35.2, −5.29)** **BDE-100 (β = −13.5, 95% CI: −26.8, −0.22)** **∑PBDEs (β = −19.0, 95% CI: −35.7, −2.37)**
Guo (2018) [30]	BDE-47, BDE-183	China	Case–control	Fifth graders from South China (n = 174)	Serum	**BDE-47 (β = −8.1, *p* < 0.05, 95% CI: −15, −1.6)** **BDE-183 (β = −4.4, *p* < 0.05, 95% CI: −8.4, −0.42)**
Chen (2018) [57]	BDE-209	China	Cross-sectional	Occupational workers from a deca-BDE manufacturing plant	Serum and urine	**BDE-209 (β = 8.632, *p* = 0.029 95% CI 0.930, 16.33)**
Zheng (2017) [61]	BDE-153, Total BDE-7, BDE-99	China	Cohort	Pregnant women (n = 72)	Women and cord serum	**BDE-153 (r = −0.494, *p* = 0.002)** **Total BDE-7 (r = −0.455 *p* = 0.004)** **BDE-99 (r = −0.365, *p* = 0.029)**
Zheng (2017) [62]	BDE-66, BDE-85	China	Cohort	Occupational e-waste recycling workers (n = 79)	Serum	**BDE-85 (β = 0.154, *p* = 0.014, 95% CI: 0.033, 0.276)** **BDE-66 (β = 0.161, *p* = 0.013, 95% CI: 0.035, 0.286)**
Preston (2017) [60]	DPHP	USA	Cohort	Office workers from the Boston area (n = 51)	Serum and urine	**DPHP (β = 0.43, 95% CI: 0.15, 0.72)**
Liu (2017) [31]	OH-BDE-47	China	Case–control	Thyroid cancer patients (n = 33)	Serum	**OH-BDE-47 (β = −2.49, 95% CI: −4.19, −0.78)**
Ding (2017) [19]	BDE-99, ∑PBDEs (−47, −99, −100, −153)	China	Cohort	Pregnant women in rural northern China (n = 107)	Cord blood	**BDE-99 (β = 0.41, 95% CI: 0.10, 0.72)** **∑PBDEs (β = 0.37, 95% CI: 0.06, 0.68)**
Makey (2016) [67]	BDE-47	USA	Cohort	Healthy adult office workers in Boston (n = 52)	Serum	**BDE-47 (β = −2.6, *p* = 0.02, 95% CI: −4.7, −0.35)**
Vuong (2015) [23]	BDE-28, BDE-47	USA	Cohort	Pregnant women from the HOME study (n = 389)	Women and cord serum	**BDE-28 (β = 0.85, *p* < 0.05, 95% CI: 0.05, 1.64)** **BDE-47 (β = 0.82, *p* < 0.05, 95% CI: 0.12, 1.51)**
Abdelouahab (2013) [16]	PBDE-47, PBDE-99, ∑PBDEs	Canada	Cohort	Pregnant women without thyroid disease (n = 260)	Serum	**PBDE-47 (β = −0.29, 95% CI: −0.51, −0.08)** **PBDE-99 (β = −0.35, 95% CI: −0.57, −0.12)** **∑PBDEs (β = −0.36, 95% CI: −0.56, −0.13)**
Stapleton (2011) [22]	∑PBDEs (−47, −99, −100) BDE-153, 4’OH-BDE-49/6-OH-BDE-47	USA	Case–control	Pregnant women >34 weeks into pregnancy (n = 137)	Serum	**∑PBDEs (−47, −99, −100) (rs = 0.20, *p* < 0.05)** **BDE-153 (rs = 0.20, *p* < 0.05)** **4’OH-BDE-49/6-OH-BDE-47 (rs = 0.18, *p* < 0.05)**
Wang (2010) [76]	BDE-126, BDE-205	China	Case–control	People exposed to an e-waste site (n = 325)	Serum	**BDE-126 (β = 0.25, Std Err= 0.10 *p* = 0.018** **BDE-205 (β = 3.27, Std Err= 0.97 *p* = 0.001**
Turyk (2008) [37]	∑PBDEs	USA	Cohort	Adult male sport fish consumers (n = 354)	Urinary and serum	**∑PBDEs (r = 0.12, *p* = 0.03)**
Herbstman (2008) [34]	BDE-100, BDE-153	USA	Cohort	Infants delivered at Johns Hopkins Hospital (n = 297)	Serum	**BDE-100 (β = 2.14, 95% CI: 1.10, 4.18)** **BDE-153 (β = 2.25, 95% CI: 1.07, 4.75)**
Julander (2005) [35]	BDE-28, BDE-100	Sweden	Cohort	Personnel working with electronic dismantling (n = 19)	Serum	**BDE-28 (r = 0.58, *p* = 0.029)** **BDE-100 (r = 0.70, *p* = 0.006)**

Bolded studies indicate statistically significant findings.

**Table 5 biomedicines-12-01365-t005:** Flame retardant and fT3.

Author (Year), Ref	Flame Retardant Type	Country	Study Design	Investigated Population (n)	Measurement of Exposure	Association with fT3
Babichuk (2023) [38]	PBB- 153, PBDE-28, PBDE-47, PBDE-99, PBDE-100, PBDE-153	Canada	Cohort	Two rural coastal populations (n = 80)	Serum	PBB-153 (β = 0.249, *p* = 0.368, 95% CI: −0.128, −0.340) PBDE-28 (β = 0.193, *p* = 0.514, 95% CI: −0.292, 0.577) PBDE-47 (β = 0.041, *p* = 0.917, 95% CI: −0.041, 0.046) PBDE-99 (β = −0.086, *p* = 0.752, 95% CI: −0167, 0.121) PBDE-100 (β = −0.249, *p* = 0.248, 95% CI: −0.056, 0.015) **PBDE-153 (β = 0.293, *p* < 0.088, 95% CI: −0.001, 0.013)**
Liu (2022) [41]	PBT, DBDPE, TCEP, TPP, EHDP	China	Case–control	Patients with or without thyroid cancer (n = 481)	Serum	PBT (β = −0.06, *p* > 0.05, 95% CI: −2.17, 2.08) DBDPE (β = 0.00, *p* > 0.05, 95% CI: −1.34, 1.27) TCEP (β = −2.00, *p* > 0.05, 95% CI: −4.21, 0.30) **TPP (β = 3.05, *p* < 0.01, 95% CI: 0.82, 5.40)** **EHDP (β = −5.34, *p* < 0.05, 95% CI: 0.60, 4.50)**
Zhao (2021) [44]	BDE-47, BDE-99, BDE-100, BDE-209	China	Case–control	Residents of a well-known FR production region (n = 172)	Serum	**BDE-47 (β = 0.082, 95% CI: 0.010, 0.155)** **BDE-100 (β = 0.063, 95% CI: 0.003, 0.123)** **BDE-99 (β = 0.057, 95% CI: 0.007, 0.107)** **BDE-209 (β = 0.037, 95% CI: (−0.010, 0.084)**
Percy (2021) [21]	BDCIPP	USA	Cohort	Pregnant women and their newborns (n = 298)	Urinary and cord serum	**BDCIPP Q2 (β = 0.04 95% CI: −0.07, 0.15)** **BDCIPP Q3 (β = −0.07 95% CI: −0.19, 0.04)** **BDCIPP Q4 (β = −0.11 95% CI: −0.22, 0.00)**
Hu (2021) [42]	BDE-28	China	Cohort	Rural adult residents along Yangtze River (n = 329)	Serum and urine	**BDE-28 (β = −0.03, *p* < 0.05, 95% CI: −0.05, −0.01)**
Zhao (2020) [50]	DBDPE	China	Case–control	DBDPE manufacturing workers (n = 104)	Hair and nail and serum	**DBDPE (r = 0.255, *p* = 0.007)**
Gravel (2020) [46]	tb-DPHP, BDE-209, BDE-47, BDCIPP, BDE-153	Canada	Cross-sectional	Electronic waste recycling workers (n = 100)	Plasma and urine	b-DPHP: (β = −0.079, *p* > 0.05, 95% CI: −0.429, 0.271) **BDE-209: (β = −0.052, *p* = 0.03, 95% CI: −0.099, −0.005)** BDE-47: β = 0.054, *p* = 0.06, 95% CI: −0.002, 0.111) BDCIPP: β = 0.090, *p* > 0.05, 95% CI: −0.063, 0.242) **BDE-153: β = −0.156, *p* = 0.01, 95% CI: −0.265, −0.047)**
Guo (2019) [54]	BDE-47, BDE-207	China	Case–control	Residents of an e-waste region (n = 112)	Serum	**BDE-47 (β = 0.15, 95% CI: 0.0036, 0.30)** **BDE-85 (β = −0.083, 95% CI: −0.16, −0.0033)**
Curtis (2019) [53]	PBBs	USA	Cohort	Children exposed to PBEs from Michigan PBB registry	Serum	**PBB (r = 3.01, *p* = 0.002)**
Vuong (2018) [59]	∑PBDEs (−28, −47, −99, −100,−153)	USA	Cohort	Mother–child pairs (n = 162)	Serum	**∑PBDEs (β = 0.25, 95% CI: 0.07, 0.43)**
Byrne (2018) [29]	∑PBDEs (−28, −33), BDE-100	USA	Case–control	Remote Alaska Native population (n = 85)	Serum	**∑PBDEs (β = 0.18, *p* < 0.005, 95% CI: 0.07, 0.30)** **BD-100 (β = 0.39, *p* < 0.005, 95% CI: 0.12, 0.66)**
Xu (2015) [68]	∑PBDEs	China	Case–control	Residents of an e-waste dismantling area in Zhejiang (n = 55)	Serum	∑PBDEs (r = −0.14, *p* = 0.317)
Vuong (2015) [23]	BDE-28, BDE-47	USA	Cohort	Pregnant women from the HOME study (n = 389)	Women and cord serum	**BDE-28 (β = 0.14, *p* < 0.05, 95% CI: 0.02, 0.26)** **BDE-47 (β = 0.12, *p* < 0.05, 95% CI: 0.01, 0.22)**
Zheng (2017) [61]	BDE-66, BDE-85	China	Cohort	Occupational e-waste recycling workers (n = 79)	Serum	**BDE-66 (β = 0.070, *p* = 0.033, 95% CI: 0.006, 0.135)** **BDE-85 (β = 0.115, *p* = 0.011, 95% CI: 0.028, 0.203)**
Kim (2013) [72]	∑PBDEs	Korea	Cohort	Pregnant women in Korea (n = 138)	Serum	**∑PBDEs (β = −0.049, *p* < 0.05, 95% CI: −0.088, −0.009)**
Abdelouahab (2013) [16]	PBDE-99, ∑PBDEs	Canada	Cohort	Pregnant women without thyroid disease (n = 260)	Serum	**PBDE-99 (β = 0.08, 95% CI: 0.03, 0.13)** **∑PBDEs (β = 0.05, 95% CI: −0.001, 0.09)**
Lin (2011) [75]	BDE-99, BDE-154, BDE-183, ∑PBDEs	Taiwan	Cohort	Mothers and their nursing infants with PBDE exposure (n = 54)	Serum	**BDE-99 (r = −0.384 *p* = 0.005)** **BDE-154 (r = −0.305 *p* = 0.026)** **BDE-183 (r = −0.271 *p* = 0.049)** **∑PBDEs (r = 0.281 *p* = 0.041)**
Wang (2010) [76]	PBB-103	China	Case–control	People exposed to an e-waste site (n = 325)	Serum	**PBB-103 (β = −0.12, Std Err= 0.05 *p* = 0.010**
Bahn (1980) [32]	PBB	USA	Cohort	Workers from a PBB manufacturing plant (n = 86)	Serum	PBB (*p* = 0.06)

Bolded studies indicate statistically significant findings.

**Table 6 biomedicines-12-01365-t006:** Flame retardant and fT4.

Author (Year), Ref.	Flame Retardant Type	Country	Study Design	Investigated Population (n)	Measurement of Exposure	Association with fT4
Babichuk (2023) [38]	PBB-153, PBDE-28, PBDE-47, PBDE-99, PBDE-100, PBDE-153	Canada	Cohort	Two rural coastal Populations (n = 80)	Serum	PBB-153 (β = −0.237, *p* = 0.424, 95% CI: −1.122, 0.478) PBDE-28 (β = 0.027, *p* = 0.933, 95% CI: −1.422, 1.548) PBDE-47 (β = −0.081, *p* = 0.849, 95% CI: −0.163, 0.135) PBDE-99 (β = −0.154, *p* = 0.597, 95% CI: −0.625, 0.362) PBDE-100 (β = 0.061, *p* = 0.790, 95% CI: −0.105, 0.138) PBDE-153 (β = −0.014, *p* = 0.939, 95% CI: −0.025, 0.023)
Liu (2022) [41]	PBT, DBDPE, TCEP, TPP, EHDP	China	Case–control	Patients with or without thyroid cancer (n = 481)	Serum	PBT (β = −0.84, *p* > 0.05, 95% CI: −3.54, 2.02) DBDPE (β = −0.89, *p* > 0.05, 95% CI: −2.58, 0.83) **TCEP (β = −4.59, *p* < 0.01, 95% CI: −7.39, −1.76)** TPP (β = 2.68, *p* > 0.05, 95% CI: −0.30, 5.76) EHDP (β = 2.12, *p* > 0.05, 95% CI: −0.49, 4.74)
Yang (2021) [43]	BDE-47	China	Cross-sectional	Patients with abnormal thyroid hormone levels (n = 40)	Serum	**BDE-47 (β = −0.37, *p* < 0.05, 95% CI: −1.69, 0.95)**
Percy (2021) [21]	BDCIPP	USA	Cohort	Pregnant women and their newborns (n = 298)	Urinary and cord serum	**BDCIPP Q2 (β = 0.02,95% CI: −0.02, 0.06)** **BDCIPP Q3 (β = 0.02, 95% CI: −0.03, 0.06)** **BDCIPP Q4 (β = −0.03, 95% CI: −0.07, 0.01)**
Hu (2021) [42]	BDE-28, BDE-47, BDE-99, BDE-100, BDE-183	China	Cohort	Rural adult residents along Yangtze River (n = 329)	Serum and urine	**BDE-28 (β = −0.05, *p* < 0.05, 95% CI:−0.08, −0.02)** **BDE-47 (β = −0.02, *p* < 0.05, 95% CI:−0.04, −0.01)** **BDE-99 (β = −0.02, *p* < 0.05, 95% CI:−0.05, −0.01)** **BDE-100 (β = −0.03, *p* < 0.05, 95% CI:−0.06, −0.01)** **BDE-183 (β = −0.03, *p* < 0.05, 95% CI:−0.06, −0.01)**
Gravel (2020) [46]	tb-DPHP, BDE-209, BDE-47, BDE-153	Canada	Cross-sectional	Electronic waste recycling workers (n = 100)	Plasma and urine	tb-DPHP: β = −0.434, *p* > 0.05, 95% CI: −3.295, 2.427 BDE-209: β = −0.001, *p* > 0.05, 95% CI: −0.191, 0.189 BDE-47: β = −0.130, *p* > 0.05, 95% CI: −0.358, 0.099 BDE-153: β = 0.248, *p* > 0.05, 95% CI: −0.196, 0.691
Guo (2019) [54]	BDE-153, BDE-183, BDE-204, BDE-207,	China	Case–control	Residents of an e-waste region (n = 112)	Serum	**BDE-153 (β = −0.62, 95% CI: −1.2, −0.083)** **BDE-183 (β = −0.063, 95% CI: −1.1, −0.16)** **BDE-204 (β = −0.59, 95% CI: −1.1, −0.092)** **BDE-207 (β = −0.50, 95% CI: −0.86, −0.14)**
Chen (2019) [51]	DBDPE	China	Case–control	Adults in a DBDPE manufacturing area (n = 302)	Serum	**DBDPE (β = 0.212, 95% CI: 0.011, 0.412)**
Guo (2018) [30]	BDE-47, BDE-99, BDE-100, BDE-183, BDE-204	China	Case–control	Fifth graders from South China (n = 174)	Serum	**BDE-47 (β = −0.82 , *p* < 0.05, 95% CI: −1.4 −0.28)** **BDE-99 (β = −0.54, *p* < 0.05, 95% CI: −0.91, −0.16)** **BDE-100 (β = −4.4 , *p* < 0.05, 95% CI: −1.0, −0.31)** **BDE-183 (β = −4.4 , *p* < 0.05, 95% CI: −0.73, −0.069)** **BDE-204 (β = −4.4 , *p* < 0.05, 95% CI: −0.39, −0.048)**
Albert (2018) [56]	BDE-47	Canada	Cohort	Healthy young men (n = 47)	Serum	**BDE-47 (β = 0.98, 95% CI: 0.02, 1.94, *p* = 0.05)**
Makey (2016) [67]	BDE-153	USA	Cohort	Healthy adult office workers in Boston (n = 52)	Serum	**BDE-153 (β = 0.35, *p* = 0.04, 95% CI: 0.03, 0.67)**
Vuong (2015) [23]	BDE-28, BDE-47	USA	Cohort	Pregnant women from the HOME study (n = 389)	Women and cord serum	**BDE-28 (β = 0.05, *p* < 0.05, 95% CI: 0.01, 0.09)** **BDE-47 (β = 0.04, *p* < 0.05, 95% CI: 0.004, 0.07)**
Xu (2015) [68]	∑PBDEs	China	Case–control	Residents of an e-waste dismantling area in Zhejiang (n = 55)	Serum	∑PBDEs (r = −0.17, *p* = 0.225)
Zheng (2017) [61]	BDE-66, BDE-85	China	Cohort	Occupational e-waste recycling workers (n = 79)	Serum	**BDE-66 (β = 0.106, *p* = 0.016, 95% CI: 0.021, 0.190)** **BDE-85 (β = 0.139, *p* = 0.022, 95% CI: 0.021, 0.258)**
Johnson (2013) [71]	PentaBDE = ∑BDE-47, BDE-99, BDE-100 OctaBDE = ∑BDE-183 and BDE-201	USA	Cohort	Men recruited from Massachusetts General Hospital (n = 62)	Serum	**PentaBDE (β = 3.6, *p* = 0.02, 95% CI: 0.6, 6.5)** **OctaBDE (β = 3.3, *p* = 0.01, 95% CI: 1.0, 5.6)**
Kim (2013) [72]	∑PBDEs	Korea	Cohort	Pregnant women in Korea (n = 138)	Serum	**∑PBDEs (β = 0.058, *p* < 0.05, 95% CI: 0.016, 0.100)**
Abdelouahab (2013) [16]	PBDE-47, PBDE-99, ∑PBDEs	Canada	Cohort	Pregnant women without thyroid disease (n = 260)	Serum	**PBDE-47 (β = 0.25, 95% CI: 0.06, 0.44)** **PBDE-99 (β = 0.27, 95% CI: 0.07, 0.46)** **∑PBDEs (β = 0.29, 95% CI: 0.09, 0.48)**
Kim (2012) [73]	BDE-49, BDE-153	South Korea	Case–control	Children with congenital hypothyroidism and their mothers (n = 76)	Serum	**BDE-49 (r = 0.584, *p* < 0.05)—Mother** **BDE-153 (r = 0.405, *p* < 0.05)—Babies**
Leijs (2012) [74]	BDE-99	Netherlands	Cohort	14–19-year-old children from Amsterdam/Zaandam region (n = 33)	Serum	**BDE-99 (*p* = 0.048)**
Stapleton (2011) [22]	∑PBDEs (−47, −99, −100), BDE-153, 4′OH-BDE-49/6-OH-BDE-47	USA	Case–control	Pregnant women >34 weeks into pregnancy (n = 137)	Serum	**∑PBDEs (−47, −99, −100) (r = 0.19, *p* < 0.05)** **4′OH-BDE-49/6-OH-BDE-47 (r = 0.17, *p* < 0.05)**
Lin (2011) [75]	BDE-99, BDE-183	Taiwan	Cohort	Mothers and their nursing infants with PBDE exposure (n = 54)	Serum	**BDE-99 (r = −0.342 *p* = 0.012)** **BDE-183 (r = −0.273 *p* = 0.048)**
Wang (2010) [76]	PBB 103	China	Case–control	People exposed to an e-waste site (n = 325)	Serum	**PBB-103 (β= −0.05, SE= 0.02 *p* = 0.003)**
Dallaire (2009) [33]	BDE-47, BDE-153	Canada	Cross-sectional	Inuit adults (n = 623)	Serum	**BDE-47 (GM = 2.16, CI: 1.84, 2.54)** **BDE-153 (GM = 2.05, CI: 1.85–2.27)**
Turyk (2008) [37]	∑PBDEs	USA	Cohort	Adult male sport fish consumers (n = 354)	Serum and urine	**∑PBDEs (r = 0.16, *p* = 0.005)**
Bahn (1980) [32]	PBB	USA	Cohort	Workers from a PBB manufacturing plant (n = 86)	Serum	PBB (*p* = 0.11)

Bolded studies indicate statistically significant findings.

**Table 7 biomedicines-12-01365-t007:** Flame retardant and thyroid cancer.

Author (Year), Ref.	Flame Retardant Type	Country	Study Design	Investigated Population (n)	Measurement of Exposure	Association with Cancer
Liu (2022) [41]	PBB, PBT, HBB, EHTBB, BTBPE, DBDPE, TPrP, TBP, TCEP, TCPP, TDCPP, TBEP, TPP, EHDP	China	Case–control	Patients with or without thyroid cancer (n = 481)	Serum	**PBB (OR = 3.19, *p* < 0.001, 95% CI: 1.79, 5.68)** **PBT (OR = 0.88 *p* < 0.001, 95% CI: 0.52, 1.48)** HBB (OR = 0.80, *p* = 0.359, 95% CI: 0.49, 1.29) **EHTBB (OR = 0.53, *p* = 0.081, 95% CI: 0.26, 1.08)** **BTBPE (OR = 0.35, *p* < 0.001, 95% CI:0.22–0.55)** DBTPE (OR = 0.97, *p* = 1.82, 95% CI:0.55–1.69) **TPrP (OR = 6.51, *p* < 0.001, 95% CI: 4.11, 10.31)** **TCEP (OR = 0.44, *p* < 0.001, 95% CI: 0.26, 0.76)** **TCPP (OR = 10.09, *p* < 0.001, 95% CI: 6.13, 16.59)** **TDCPP (OR = 2.11, *p* < 0.001, 95% CI: 2.11, 5.08)** **TBEP (OR = 6.37, *p* < 0.001, 95% CI: 4.02, 10.10)** TPP (OR = 0.72, *p* = 0.255, 95% CI: 0.41, 1.26) **EHDP (OR = 0.01, *p* < 0.001, 95% CI: 0.00, 0.02)**
Zhang (2021) [28]	BDE-209, ∑8 PBDEs = PBDE-28, PBDE-47, PBDE-99, PBDE-100, PBDE-153, PBDE-154, PBDE-183, PBDE-209, BDE-28, BDE-47, BDE-183 BDE-209, ∑7 PBDEs = 28, −47, −99, −100, −153, −154, −183	China	Case–control	Thyroid cancer patients from Anhui province (n = 616)	Fasting blood	**BDE-209 (OR =0.31, 95% CI: 0.15, 0.66, *p* = 0.002)—Pb and BDE** **∑8 PBDEs (OR =0.33, 95% CI: 0.15, 0.69, *p* = 0.003)—Pb and BDE** **BDE-28 (OR =3.17, 95% CI: 1.49, 6.78, *p* = 0.002)—Hg and BDE** **BDE-47 (OR =2.58, 95% CI: 1.19, 5.59, *p* = 0.01)—Hg and BDE** **BDE-183 (OR =0.34, 95% CI: 0.16, 0.75, *p* = 0.01)—Hg and BDE** **BDE-209 (OR =3.67, 95% CI: 1.72, 7.83, *p* = 0.001)—Hg and BDE** **∑7 PBDEs (OR =2.44, 95% CI: 1.13, 5.25, *p* = 0.03)—Hg and BDE** **∑8 PBDEs (OR =4.01, 95% CI: 1.87, 8.64, *p* < 0.001)—Hg and BDE**
Huang (2020) [47]	BDE-28	USA	Case–control	US military personnel (n = 148)	Serum during active duty	**BDE-28 (OR = 2.09, 95% CI: 1.05, 4.15, *p* = 0.02)** **BDE-28 (OR = 10.74, 95% CI: 1.93, 59.72, *p* = 0.0054)—Females**
Kassotis (2020) [48]	TCEP, TDCIPP, 4-tBPDPP, B4tBPPP, T4tBPP, DiNP, TOTM, BDE-100	USA	Case–control	Adults in central North Carolina (n = 72)	Chemical mixtures isolated from personal silicone wristband samplers	**TCEP (OR = 2.3, 95% CI: 1.02, 5.05)** **TDCIPP (OR = 3.5, 95% CI: 1.20, 10.47)** **4–5BPDPP (OR = 4.6, 95% CI: 1.67, 12.71)** **B4tBPPP (OR = 5.6, 95% CI: 2.03, 15.34)** **T4tBPP (OR = 3.6, 95% CI: 1.74, 7.37)** **DiNP (OR = 9.5, 95% CI: 2.37, 38.29)** **TOTM (OR = 3.5, 95% CI: 1.09, 11.15)** **BDE-100 (OR = 0.5, 95% CI: 0.30, 0.94)**
Deziel (2019) [27]	BDE-209	USA	Case–control	Thyroid cancer population in Connecticut (n = 500)	Serum	**BDE-209 (OR = 0.47, 95% CI: 0.23, 0.98, *p* < 0.05)**
Hoffman (2017) [2]	BDE-209, TCEP	USA	Case–control	Patients with papillary thyroid cancer at Duke University Hospital (n = 140)	Serum	**BDE-209 (OR = 2.29, 95% CI: 1.03, 5.08, *p* = 0.04)** **TCEP (OR = 2.42 CI: 1.10, 5.33, *p* = 0.03)**
Aschebrook-Kilfoy (2015) [66]	BDE-47, BDE-99, BDE-100, BDE-153	USA	Case–control	Nested CC in thyroid, prostate, lung, colorectal, and ovarian cancer screening trial (n = 311)	Mass spectrometry of serum	∑PBDEs (OR = 0.94, 95% CI: 0.79, 1.11)—All Thyroid Cancer BDE-47 (OR = 0.95, 95% CI: 0.80, 1.12)—All Thyroid Cancer BDE-99 (OR= 0.95, 95% CI: 0.81, 1.11)—All Thyroid Cancer BDE-100 (OR = 0.96, 95% CI: 0.84, 1.09)—All Thyroid Cancer BDE-153 (OR = 0.96, 95% CI: 0.82, 1.11)—All Thyroid Cancer ∑PBDEs (OR = 0.96, 95% CI: 0.79, 1.17)—Papillary Thyroid Cancer BDE-47 (OR = 0.99, 95% CI: 0.81, 1.20)—Papillary Thyroid Cancer BDE-99 (OR = 0.97, 95% CI: 0.81, 1.16)—Papillary Thyroid Cancer BDE-100 OR = 1.00, 95% CI: 0.86, 1.16)—Papillary Thyroid Cancer BDE-153 (OR = 0.96, 95% CI: 0.79, 1.15)—Papillary Thyroid Cancer

Bolded studies indicate statistically significant findings.

**Table 8 biomedicines-12-01365-t008:** Flame retardant and thyroid-related antibodies.

Author (Year), Ref.	Flame Retardant Type	Country	Study Design	Investigated Population (n)	Measurement of Exposure	Association with TPO-Ab
Zhao (2021) [44]	BDE-99	China	Case–control	Residents of a well-known FR production region (n = 172)	Serum	**BDE-99 (β = 0.085, 95% CI: 0.030, 0.140)**
Chen (2019) [51]	DBDPE	China	Case–control	Adults in a DBDPE manufacturing area (n = 302)	Serum	**DBDPE (β = 0.038, 95% CI: 0.019, 0.058)**

Bolded studies indicate statistically significant findings.

## Data Availability

The original data presented in the study is available in the original publications.

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
