# Peer review of "A Review of the Association between Exposure to Flame Retardants and Thyroid Function"

_biomedicines, 2024, doi:10.3390/biomedicines12061365_

Round 1

Reviewer 1 Report

Comments and Suggestions for Authors

The study addresses an important public health concern by examining the association between flame retardants and thyroid function, which is relevant given the widespread use of these chemicals and their known endocrine disrupting properties. The paper is commendable for its comprehensive review of an extensive literature. However, there are some suggested revisions that could improve its overall impact.

Ÿ   In the Introduction section, it would be beneficial to include results and references that show how exposure to flame retardants has changed over time. This historical perspective could provide valuable context for understanding current exposure levels and regulatory changes.

Ÿ   The discussion of flame retardant bans (lines 324-352) could be simplified and moved to the introduction. In addition, it would be informative to include the reasons for banning certain flame retardants.

Ÿ   As mentioned in the introduction, the effect of flame retardants on thyroid hormones may vary with concentration (Reference 9). Therefore, it would be beneficial to discuss the exposure levels or concentrations in each study to provide a clearer understanding of their effects. It would be useful to include specific numerical concentration data or at least information on whether the exposure is at a very high concentration due to occupational exposure or accidents.

Ÿ   Everyone is affected by flame retardants, including pregnant women and their fetuses (References 10-20). It is important to identify which populations are more vulnerable to these effects is crucial. It would be beneficial if the results section included an analysis and description of how different populations, particularly vulnerable ones, are affected by exposure to flame retardants.

Ÿ   In the introduction, it would be beneficial to provide an explicit statement of the objectives or questions addressed by the review. For example, this review not only examines the relationship between flame retardants and thyroid hormone levels, but also analyzes their relationship with cancer. This broader scope should be clearly stated at the outset to help the reader understand the comprehensive approach of the study.

Ÿ   Is this paper a simple review or a systematic review? If it is intended to be a systematic review, it should be stated in the title and the paper should be revised and improved according to the PRISMA checklist. Based on my assessment, this does not appear to be a systematic review. To avoid confusion among readers, please revise the term "systematic research" in the abstract to more accurately reflect the type of review conducted.

Ÿ   In the methods section, it is important to specify the inclusion and exclusion criteria for the review. The flow diagram (Fig. 1) shows that some studies were excluded due to “incorrect or unspecified outcome variables” and “missing statistical analysis”. These criteria should be clearly described in the methods to ensure transparency and reproducibility of the review process.

Ÿ   The inclusion criteria state that studies must "provide measurement of serum or plasma flame retardants," yet Table 1 includes cases where measurements were taken from urine. This discrepancy suggests a potential study selection problem. It is important to clarify whether urine-based measurements were considered acceptable under the inclusion criteria or whether there was an oversight in the selection process that needs to be corrected.

Ÿ   In the Methods section, the description of the data collected regarding anti-thyroid autoantibodies and cancer needs to be more detailed.

Ÿ   The results section is detailed and clearly presents both positive and negative associations. However, it is important to provide insight into why different studies show different associations (positive, negative, or no association) with flame retardants. Factors such as differences in the levels of flame retardant concentrations, age, sex, or other demographic or methodological variables could influence these results. The current content in the results section simply lists these findings without synthesizing them into a coherent conclusion. A more in depth analysis or discussion that explains these variations is needed to draw meaningful conclusions from the data presented.

Ÿ   When presenting results, it is better to reflect the clinical significance of each hormone and antibody. Since most thyroid hormones in the blood are thyroxine (T4) rather than triiodothyronine (T3), and free T4 has more clinical relevance than total T3 or free T3, I recommend reordering the results section to reflect this clinical importance. The suggested order would be TSH → free T4 → free T3 → total T3. Additionally, since thyroid autoantibodies are related to thyroid function, including them in the sequence before discussing thyroid cancer would provide a logical flow. Therefore, an optimal order for presenting the results might be TSH → free T4 → free T3 → total T3 → thyroid autoantibody → thyroid cancer. This arrangement would better align the results with their clinical relevance and importance in diagnosing and understanding thyroid disorders.

Ÿ   In Table 7, the entries for the studies by Zhang (2021) and Deziel (2019) are missing information on the number of participants involved.

Ÿ   The presence of anti- TPO antibodies is primarily observed in Hashimoto's thyroiditis and postpartum thyroiditis, but not typically associated with Graves' disease.

Ÿ   The author mentioned, "Several studies have noted the association between flame retardants and elevated thyroid-related antibody concentrations." If only two studies are being referred to here, it would be more accurate to specify this. The term "several" generally implies a larger number, which could potentially mislead readers about the breadth of evidence supporting this association. Clarifying this in the text to accurately reflect the number of studies—such as stating "two studies" instead of "several"—would improve the precision and transparency of the information provided.

Ÿ   The discussions might be strengthened by providing specific recommendations for future research.

Author Response

Reviewer 1:

The study addresses an important public health concern by examining the association between flame retardants and thyroid function, which is relevant given the widespread use of these chemicals and their known endocrine disrupting properties. The paper is commendable for its comprehensive review of an extensive literature. However, there are some suggested revisions that could improve its overall impact.

We want to thank the reviewer for providing a comprehensive review of our manuscript and recommendations to improve the study. Below, you will find our point-by-point answers to the comments raised by the reviewer. The provided line numbering refers to the clean version of the revised manuscript. 

In the Introduction section, it would be beneficial to include results and references that show how exposure to flame retardants has changed over time. This historical perspective could provide valuable context for understanding current exposure levels and regulatory changes.

Thank you for these timely edits. We have included citations that support the historical changes of the use and regulation of flame retardants overtime in lines 74-88. Please see our changes in the following paragraph:

The rising health concerns and regulatory response of flame retardants and their implications on human health has significantly changed overtime. While these molecules have been around for centuries, there has been a significant rise in their use in the 20th century4. In 1953, the United States passed the Flammable Fabrics Act, which required the use of flame retardants in many children products like clothing and interior furnishings like carpets and rugs9.  In 1975, California legislators passed the flammability standard, TB-117, which required more stringent implementation of flammability standards to ensure furniture safety from hazards of ignition10. However, in 1977, the earliest flame retardants, polychlorinated biphenyls (PCBs), were banned in the United States for their significant toxic effects on endocrine disruption, cancer, liver damage, and neurodevelopmental abnormalities. Industries switched from chlorinated flame retardants to the use of brominated flame retardants10. After noting the toxicity and persistence of these effects, with additional recognition of reproductive implications and neurobehavioral effects on children, The European Union took further measures to ban several PBDEs in 20086.   

We have also discussed the transition to more recent legislative interventions in the following paragraph of the introduction in lines 90-120.

More recently, several countries have taken measures to ban exposure to flame retardants. Countries in the European Union spearheaded efforts to ban the use of brominated flame retardants within electronics, furniture, and other products due to a combination of environmental concerns from persistence and permeability in the environment and human health effects such as endocrine disruption, neurological interference, reproductive issues, and potential carcinogenic properties 11. The United States Consumer Product Safety Commission has also banned the sales of certain products that contain PBDEs for similar reasons as prior legislation with a particular emphasis on vulnerable populations including infants and children 12. Many flame retardants, including tetrabrominated diphenyl ethers, hexabrominated diphenyl ethers, decabromodiphenyl ether, and hexabromocyclododecane, have been listed under several legislations, including the Stockholm Convention on Persistent Organic Pollutants13. Only several states within the United States including New York and California, have begun passing bills against products containing these molecules 14. New York became the first state in the nation to restrict the use of halogenated flame retardants in 2021 within electronics. Washington State has also released reports on the detriments of certain flame retardants to vulnerable populations, including firefighters 14. The Safer States organization released a statement titled “Sign on Letter to Textile Certifiers” addressing polybrominated and polymeric flame retardants that substantiated concerns about the continued use of flame retardants. This bill served as a catalyst for enacting stricter legislation, with a recommendation to expand analyses of preexisting flammability standards to a broader set of products 14.

The discussion of flame retardant bans (lines 324-352) could be simplified and moved to the introduction. In addition, it would be informative to include the reasons for banning certain flame retardants.

Thank you for this comment. This section was moved from the discussion to the introduction and further simplified. We addressed the banning of flame retardants within the chronological progression of how regulations on flame retardants changed over time. In line 82-120, we addressed, to the best of our abilities, the reasons why certain classes of flame retardants were banned including the primary concerns of certain legislations. The majority of the reasons addressed both environmental concerns of these molecules as well as the toxic physiological effects on humans including endocrine disruption, neurological effects, and potential carcinogenic effects. The relevant text is included below:

“However, in 1977, the earliest flame retardants, polychlorinated biphenyls (PCBs), were banned in the United States for their significant toxic effects on endocrine disruption, cancer, liver damage, and neurodevelopmental abnormalities. Industries switched from chlorinated flame retardants to the use of brominated flame retardants10. After noting the toxicity and persistence of these effects, with additional recognition of reproductive implications and neurobehavioral effects on children, The European Union took further measures to ban several PBDEs in 20086.   

More recently, several countries have taken measures to ban exposure to flame retardants. Countries in the European Union spearheaded efforts to ban the use of brominated flame retardants within electronics, furniture, and other products due to a combination of environmental concerns from persistence and permeability in the environment and human health effects such as endocrine disruption, neurological interference, reproductive issues, and potential carcinogenic properties 11. The United States Consumer Product Safety Commission has also banned the sales of certain products that contain PBDEs for similar reasons as prior legislation with a particular emphasis on vulnerable populations including infants and children 12. Many flame retardants, including tetrabrominated diphenyl ethers, hexabrominated diphenyl ethers, decabromodiphenyl ether, and hexabromocyclododecane, have been listed under several legislations, including the Stockholm Convention on Persistent Organic Pollutants13. Only several states within the United States including New York and California, have begun passing bills against products containing these molecules 14. New York became the first state in the nation to restrict the use of halogenated flame retardants in 2021 within electronics. Washington State has also released reports on the detriments of certain flame retardants to vulnerable populations, including firefighters 14. The Safer States organization released a statement titled “Sign on Letter to Textile Certifiers” addressing polybrominated and polymeric flame retardants that substantiated concerns about the continued use of flame retardants. This bill served as a catalyst for enacting stricter legislation, with a recommendation to expand analyses of preexisting flammability standards to a broader set of products 14.”

As mentioned in the introduction, the effect of flame retardants on thyroid hormones may vary with concentration (Reference 9). Therefore, it would be beneficial to discuss the exposure levels or concentrations in each study to provide a clearer understanding of their effects. It would be useful to include specific numerical concentration data or at least information on whether the exposure is at a very high concentration due to occupational exposure or accidents.

We agree with the reviewer that exposure levels in each study would strengthen our study. We have modified our Table 1 to include a final column with “Relative Exposure Level; Concentration, Distribution Frequencies.” While not all papers clearly addressed the relative concentration exposure, we have done our best to include information that is relevant to exposure concentration of certain flame retardants and identification of evidently high concentrations and their context of either occupational exposure, location, accidents. We have also included further analyses of concentration of flame retardants within our discussion section based on concentration and distribution frequencies, as well as general trends noted. However, it is difficult to compare high versus low concentrations of these molecules across different studies as different controls, populations, and demographic variables contributed to significant heterogeneity. However, we provided information regarding comparisons referenced in the literature, some of which compared countries or regions as well as the concentrations of different flame retardants.

Everyone is affected by flame retardants, including pregnant women and their fetuses (References 10-20). It is important to identify which populations are more vulnerable to these effects is crucial. It would be beneficial if the results section included an analysis and description of how different populations, particularly vulnerable ones, are affected by exposure to flame retardants.

Thank you for your comment. The organization of our results section addressed papers associated with certain thyroid hormones, with a description of papers that addressed vulnerable populations within each category. We have addressed the different populations that are affected by exposure to flame retardants and have placed greater focus on discussion on vulnerable populations and how they are affected by these molecules. More specifically, we addressed at risk populations including infants and children, pregnant women, and women, which is included in lines 1346-1375:

“Infants and children, a population in which thyroid hormone regulation is especially critical in development. Vuong et al noted that prenatal exposure to certain flame retardants including BDE-47, -99, and -100 predisposes children to not only thyroid dysregulation, but also cognitive impairment and ADHD-like behavior58. Interference with regulation by flame retardants poses an especially serious effect on this population. Not only is their exposure to these molecules relatively more dangerous than the general population, but children are also likely to be exposed to greater quantities of flame retardants. These molecules, present in dust, furniture, and electronics, are easily ingested by children. The smaller body size and higher metabolic demands of children also subjects them to greater consumption, which proportionally contains greater levels of these toxins4.

Moreover, women are at greater risk than men for the development of thyroid dysregulation and cancer29,46. Preston et al noted that the higher concentrations of DPHP has more profound effects on thyroid hormone levels in women compared to men29,46,54,59,65. Proposed reasoning of these sex-specific association is due to increased levels of estrogen, which plays a crucial role in thyroid regulation26,29,54,59,70. Additionally, women have greater sensitivity to thyroid hormone due to more frequent physiological hormone level changes through pregnancy and menstrual cycles. They also have higher rates of thyroid autoimmune conditions and thyroid cancer, making them more susceptible to thyroid disrupting factors26,29,54,59. Other factors including increased body fat percentage, resulting in greater accumulation of flame retardants54,59,70.”

Evidence for individuals living near electronic-waste dismantling sites, as well as citizens living in countries with more lenient legislation is also included in lines 1327-1341:

“Additionally, location, whether it be country, state or even region-specific differences in both concentrations and regulations of flame retardants are notable considerations. Jacobson et al noted that the concentration of PBDEs analyzed in their study were similar to that of other studies conducted in the US, an average concentration 2-3x higher than that noted in other countries like Europe and Asia62. Similarly, Lignell et al noted that the concentration in their population of Sweden pregnant women was about ten times lower body burden than comparable studies in North American women64. Zhang and colleagues analyzed legislative intervention for the flame retardant BDE-209, found in relatively higher concentrations than other flame retardants both in the environment and in humans. This study conducted in China, noted relatively higher concentrations than in the studies conducted in the United States2,29,63, likely due to the lack of regulation on this flame retardant at the time of publication. Similar extremely high relative concentration and detection frequencies were noted in Zhao and colleagues49. Only recently has China included BDE-209 in their list of contaminants under the List of Key Emerging Contaminants under Control in 202385.”

In the introduction, it would be beneficial to provide an explicit statement of the objectives or questions addressed by the review. For example, this review not only examines the relationship between flame retardants and thyroid hormone levels, but also analyzes their relationship with cancer. This broader scope should be clearly stated at the outset to help the reader understand the comprehensive approach of the study.

We agree with the reviewer that we should add a statement addressing the key points and that we not only examine the relationship between flame retardants and hormone levels, but also with thyroid cancer. We have added a sentence addressing these objectives in the introduction in lines 148-149:

“This review not only examines the relationship between flame retardants and thyroid hormone levels, but also analyzes their relationship with cancer.”

Is this paper a simple review or a systematic review? If it is intended to be a systematic review, it should be stated in the title and the paper should be revised and improved according to the PRISMA checklist. Based on my assessment, this does not appear to be a systematic review. To avoid confusion among readers, please revise the term "systematic research" in the abstract to more accurately reflect the type of review conducted.

We thank the reviewer for this suggestion. We recognize and apologize for this confusion. We have removed the term “systematic” from our abstract to more accurately represent our paper.

In the methods section, it is important to specify the inclusion and exclusion criteria for the review. The flow diagram (Fig. 1) shows that some studies were excluded due to “incorrect or unspecified outcome variables” and “missing statistical analysis”. These criteria should be clearly described in the methods to ensure transparency and reproducibility of the review process.

Thank you for your comment. An elaboration in the methods section addressing common examples of why certain papers were excluded has now been included in lines 342-345:

“Articles with incorrect or unspecified outcome variables included measurements of quantity of metals such as lead excretion or enzyme function like thyroid deiodinase activity as primary outcome variables were also excluded.”

The inclusion criteria state that studies must "provide measurement of serum or plasma flame retardants," yet Table 1 includes cases where measurements were taken from urine. This discrepancy suggests a potential study selection problem. It is important to clarify whether urine-based measurements were considered acceptable under the inclusion criteria or whether there was an oversight in the selection process that needs to be corrected.

Thank you for this thoughtful comment. Urine-based as well as placental, cord, and breast milk measurements were considered acceptable measures in papers that were included. We have modified the methods section inclusion criteria in lines 291-295 to state:

“Articles were included if they met the following inclusion criteria: (1) provided measurement of serum, plasma, cord, placental, breast milk, or urine flame-retardants, (2) measurements of serum, plasma, cord, placental, breast milk, or urine thyroid hormone (free thyroxine (fT4), total thyroxine (tT4),  free triiodothyronine (fT3), total triiodothyronine (tT3), thyroid-stimulating hormone (TSH))”.

In the Methods section, the description of the data collected regarding anti-thyroid autoantibodies and cancer needs to be more detailed.

We want to thank the reviewer for this suggestion..We provided a better description of how papers analyzing the association between flame retardant exposure and anti-thyroid autoantibodies and cancer have collected data and performed their analyses on page 6 in lines 324-333:

“The data gathered on anti-thyroid autoantibodies and thyroid cancer generally examined a population of age- and/or gender-matched healthy patients to a population of patients diagnosed with thyroid cancer. The measurements of flame retardants are often lipid-adjusted to account for the lipophilic qualities of these chemicals and standardized for differences of individual fat content. Several studies identified differences in risk of cancer development based on concentration of flame retardants (Hoffmann, Deziel 2019, Zhang 2021). The utilization of statistical analyses, most commonly logistic regression and odds ratios were used to establish this association. Patients with greater exposure to certain flame retardants were noted to have higher odds (OR>1) of developing thyroid cancer.”

The results section is detailed and clearly presents both positive and negative associations. However, it is important to provide insight into why different studies show different associations (positive, negative, or no association) with flame retardants. Factors such as differences in the levels of flame retardant concentrations, age, sex, or other demographic or methodological variables could influence these results. The current content in the results section simply lists these findings without synthesizing them into a coherent conclusion. A more in depth analysis or discussion that explains these variations is needed to draw meaningful conclusions from the data presented.

Thank you for this comment. A more detailed analysis was conducted and synthesized to draw more meaningful conclusions noted in lines 1233-1329. We addressed concentrations of flame retardants, sex, age, as well as occupational exposure, and were able to draw several additional meaningful conclusions including the increased risk of development of thyroid dysregulation and thyroid cancer in infants and children, in women compared to men, and increased risk with greater concentration exposure.

When presenting results, it is better to reflect the clinical significance of each hormone and antibody. Since most thyroid hormones in the blood are thyroxine (T4) rather than triiodothyronine (T3), and free T4 has more clinical relevance than total T3 or free T3, I recommend reordering the results section to reflect this clinical importance. The suggested order would be TSH → free T4 → free T3 → total T3. Additionally, since thyroid autoantibodies are related to thyroid function, including them in the sequence before discussing thyroid cancer would provide a logical flow. Therefore, an optimal order for presenting the results might be TSH → free T4 → free T3 → total T3 → thyroid autoantibody → thyroid cancer. This arrangement would better align the results with their clinical relevance and importance in diagnosing and understanding thyroid disorders

Thank you for his thoughtful comment. We agree with the reviewer’s point that it is better to reflect the clinical significance of each hormone and antibody. We have reorganized the order of the manuscript according to the recommendation provided to better reflect the clinical significance and logical flow of the paper.

In Table 7, the entries for the studies by Zhang (2021) and Deziel (2019) are missing information on the number of participants involved.

Thank you for this comment. We have included the number of participants for the studies by Zhang (2021) and Deziel (2019) in Table 7.

The presence of anti- TPO antibodies is primarily observed in Hashimoto's thyroiditis and postpartum thyroiditis, but not typically associated with Graves' disease.

The reviewer is correct that the presence of anti-TPO antibodies is typically not associated with Grave’s disease.Therefore,  we have removed “Graves’ disease” from this sentence discussing association of anti- TPO antibodies.

The author mentioned, "Several studies have noted the association between flame retardants and elevated thyroid-related antibody concentrations." If only two studies are being referred to here, it would be more accurate to specify this. The term "several" generally implies a larger number, which could potentially mislead readers about the breadth of evidence supporting this association. Clarifying this in the text to accurately reflect the number of studies—such as stating "two studies" instead of "several"—would improve the precision and transparency of the information provided.

We have reworded this sentence to state “two studies” instead of “several studies” to better represent the quantity of studies being addressed.

The discussions might be strengthened by providing specific recommendations for future research.

We thank the reviewer for this suggestion. The discussion now includes more specific recommendations including creating models that account for variables such as age, gender, BMI, occupational exposure, health literacy and socioeconomic determinants of health, that may impact individuals affected and diagnosed with the development of thyroid disruption or thyroid cancer. Additionally studies may benefit from surveying which patients are aware of the presence and detriments of flame retardants. Ultimately screening and risk stratifying patients to prevent long-term often irreversible sequelae of exposure to these toxins. These recommendations are included in lines 1424-1437:

“Additionally, considering the mixed correlation results of many flame retardants, future studies should place greater emphasis on the other variables that may be influencing the lack of consistent associations between flame retardants and thyroid hormone. Creating statistical models that account for variables such as age and gender, BMI, occupational exposure, but also educational awareness, and other socioeconomic determinants of health that may affect access to proper medical care and influence the ability for early detection. It is similarly important to focus on identifying at-risk populations with exposure to flame retardants, educating populations on the potential repercussions of exposure, and implementing strategies to mitigate. Moreover, future studies may also benefit from more widespread and robust detection of flame retardants within the environment and within humans. As legislation has universally been shown to be slow to adapt, studies may also benefit from surveying whether patients and physicians are aware of the detriments of flame retardants.”

Reviewer 2 Report

Comments and Suggestions for Authors

Dear authors, your work is both important and significant.

The primary concern lies in the discrepancy between the intended methodology, as stated in the study aim: "To provide an overview of the current literature and update the meta-analysis published in 2015, ...", and the actual approach employed, which was a narrative review.

Therefore, I recommend taking the final step and transforming this manuscript into a systematic review, with or without a meta-analysis. If the authors choose to proceed with a systematic review, it is advisable to expand the methodology section to provide a more detailed description of the article search and selection process. Additionally, selecting an internationally recognized appraisal tool to assess the quality of eligible articles for inclusion in the review would enhance rigor. Alternatively, if the decision is to pursue a meta-analysis, outlining the steps for data pooling is essential.

Author Response

Reviewer 2:

Dear authors, your work is both important and significant.

The primary concern lies in the discrepancy between the intended methodology, as stated in the study aim: "To provide an overview of the current literature and update the meta-analysis published in 2015, ...", and the actual approach employed, which was a narrative review.

Therefore, I recommend taking the final step and transforming this manuscript into a systematic review, with or without a meta-analysis. If the authors choose to proceed with a systematic review, it is advisable to expand the methodology section to provide a more detailed description of the article search and selection process. Additionally, selecting an internationally recognized appraisal tool to assess the quality of eligible articles for inclusion in the review would enhance rigor. Alternatively, if the decision is to pursue a meta-analysis, outlining the steps for data pooling is essential.

We want to thank the reviewer for providing recommendations to improve our study.. We do see that our statement of expanding on the prior meta-analysis may be misleading. We understand that moving forward with a systematic review would benefit the study. Our team chose not to proceed with a systematic review primarily because of the wide heterogeneity across the papers, particularly with collection of data statistics, methodologies utilized, and differences in patient populations analyzed, which made it difficult to have a reliable representation of anticipated results. 

Reviewer 3 Report

Comments and Suggestions for Authors

This article reviews studies evaluating the effects of flame retardants on thyroid function.

Comments:

1. It is recommended that the description of the known and potential mechanisms of the effects of different flame retardants on thyroid function be strengthened.

2. It would be helpful to add figures demonstrating the structural similarity of flame retardants to the thyroid hormones discussed in the text

3. In the introduction, it is recommended that a classification of flame retardants be added to make it easier for the reader to interpret the data in the tables. It is also recommended to interpret all the abbreviations used.

4. It is recommended to add references to the literature in the tables, as it is difficult to find the primary article by one name and year

Author Response

Reviewer 3:

This article reviews studies evaluating the effects of flame retardants on thyroid function.

We want to thank the reviewer for providing a comprehensive review of our manuscript and recommendations to improve our study. Below, you will find our point-by-point answers to the comments raised by the reviewer. The provided line numbering refers to the clean version of the revised manuscript.

Comments:

  1. It is recommended that the description of the known and potential mechanisms of the effects of different flame retardants on thyroid function be strengthened.

We agree with the reviewer that the mechanisms of the effects of different flame retardants on thyroid function would strengthen our study.  We have addressed the known and proposed mechanisms of impacts of different flame retardants on thyroid function more comprehensively through discussion of additional mechanisms such as effects on sulfotransferase, allosteric inhibition, more detail on specific deiodinase inhibition, intranuclear receptor effects, effects on binding affinity and half-life, as well as relationship to possible chronic exposure as a rationale for the carcinogenic effects of these molecules. New discussion has been included in lines 1131-1226 as included in the text below:

“Gravel et al noted that the hydroxylated metabolites act as competitive inhibitors that ultimately prevent gene expression45. This paper also noted that allosteric activation of several organophosphate esters, TPhP and TDCIPP, were observed. These molecules would increase the binding of free T4 to transport proteins through this mechanism, resulting in conformational changes in transport proteins45.”

Several papers also noted this as a mechanism of PBDEs in inhibiting activity of sulfotransferases, enzymes involved in the metabolism of thyroid hormone22,63,66. This study noted that PBDEs result in direct reduction of tT4 levels, and that these effects may be tissue-specific66. Leonetti et al also described how brominated flame retardants inhibit deiodinases, specifically DIO3, within the placenta63. This paper also described how flame retardants may influence sulfotransferase activity in the placenta63. The use of hormone ratios, including fT4 to tT4 and fT3 to tT3, as indicators of transport protein involvement, whereas ratios of fT4 to fT3 can be used as indicators of deiodinase involvement, an enzyme that is used in the conversion of T4 to T345

Several studies have noted intranuclear mechanisms of action from several PBDEs and metabolites acting on receptors such as thyroid hormone receptors and estrogen receptors, modulating the transcription of several genes26,70

These hydroxylated PBDEs have been found to possess stronger endocrine disruption through protein binding to molecules such as transthyretin and TBG with higher affinity resulting in shorter half-life of T4, in addition to inhibition of thyroid deiodinase26,82 and interfere with binding of thyroid hormone to human receptors75. This reaction may also explain why the analysis of flame retardants in vitro and in animal studies may differ from analyses within humans. Yet further unelucidated mechanisms are suspected, as the relationship between flame retardant exposure and carcinogenicity is thought to be regulated by additional indirect and chronic mechanisms46.

  1. It would be helpful to add figures demonstrating the structural similarity of flame retardants to the thyroid hormones discussed in the text.

We thank the reviewer for this suggestion. We created a figure through our institution library that helps display the structural similarities between thyroid hormone and flame retardants after line 52:

  1. In the introduction, it is recommended that a classification of flame retardants be added to make it easier for the reader to interpret the data in the tables. It is also recommended to interpret all the abbreviations used.

Thank you for this comment. We have added a brief section within the introduction that addresses the general classification of flame retardants that will hopefully help with interpreting the data in the tables in lines 64-72:

“Flame retardants are generally classified into several different categories including halogenated flame retardants, which includes brominated flame retardants such as polybrominated diphenyl ethers (PBDEs), hexabromocyclododecane (HBCDs), and tetrabromobisphenol (TBBPA), and chlorinated flame retardants such as Tris(2,3-dibromopropyl) phosphate (TDBPP or Tris), chlorinated tris (Tris(1,3-dichloro-2-propyl) phosphate). Addition categories include organophosphorus flame retardants including Triphenyl phosphate (TPP), Tris(2-chloroethyl) phosphate (TCEP), Tris(2-butoxyethyl) phosphate (TBEP)), aswell as other categories including nitrogen-based, inorganic, intumescent, mineral, and reactive flame retardants 4.”

We have also included an alphabetical list of abbreviations and full names of flame retardants in the appendix to help with interpreting the tables.

  1. It is recommended to add references to the literature in the tables, as it is difficult to find the primary article by one name and year

We thank the reviewer for this suggestion. We have included updated references throughout the manuscript and have included them in Table 1 to facilitate identifying the corresponding paper.

Round 2

Reviewer 2 Report

Comments and Suggestions for Authors

The authors made substantial efforts to improve the quality of manuscript.